# A comparison of the atmospheric response to the Weddell Sea Polynya in AGCMs of varying resolutions.

Holly. C. Ayres[1,2], David. Ferreira[2], Wonsun. Park[3,4,5], Joakim. Kjellsson[3,6], Malin. Ödalen[3]

[1] National Oceanography Centre, Southampton, UK

[2] Department of Meteorology, University of Reading, Reading, UK

[3] Division of Ocean Circulation and Climate Dynamics, GEOMAR Helmholtz Centre for Ocean Research Kiel, Germany

[4] IBS Center for Climate Physics, Institute for Basic Science (IBS), Busan, Republic of Korea

[5] Department of Climate System, Pusan National University, Busan, Republic of Korea

[6] Kiel University, Kiel, Germany

*Correspondence to*: Holly Ayres holly.ayres@noc.ac.uk

**Abstract.** The Weddell Sea Polynya (WSP) is a large opening within the sea ice cover of the Weddell Sea sector. It has been a rare event in the satellite period, appearing between 1973 and 1976, and again in 2016/17. Coupled modelling studies have suggested that there may be a large-scale atmospheric response to the WSP. Here, the direct atmospheric response to the WSP is estimated from atmosphere-only numerical experiments. Three different models, the HadGEM3 UK Met Office model, the ECHAM5 Max Planck Institute model and the OpenIFS ECMWF model, each at two different resolutions, are used to test the robustness of our results. The use of large ensembles reduces the weather variability and isolates the atmospheric response. Results show a large (~100-200 Wm$^{-2}$) turbulent air-sea flux anomaly above the polynya. The response to the WSP is local and of short duration (barely outlasting the WSP) with a similar magnitude and spatial pattern of lower tropospheric warming and increase in precipitation in all 6 configurations. All models show a weak decrease in surface pressure over the WSP, but this response is small (~2 hPa) in comparison to internal variability. The dynamic response is inconsistent between models and resolutions, above the boundary layer, suggesting a weak or null response that is covered by internal variability aloft. The higher resolution does not alter the pattern of the response but increases its magnitude by up ~50 % in two of the three models. The response is influenced by natural variability of the westerly jet. The models perform well against ERA5 reanalysis data for the 1974 WSP in spatial response and magnitude, showing a turbulent heat flux of approximately 150 Wm$^{-2}$.

## 1 Introduction

The Weddell Sea Polynya (WSP) is a large opening within the sea ice cover of the Weddell Sea sector, typically found over the Maud Rise (65 °S, 2.5 °E) inside the Weddell Gyre, in its largest occurrences. It has been a rare event in the satellite period, appearing between 1973 and 1976 and again smaller in 2016/7 (Swart et al., 2018; Campbell et al., 2019; Cheon and Gordon, 2019). The WSP has previously opened on the 22nd of November 1973, 13th September 2017, with the largest occurrence on the 23rd of September 1974 (Francis et al., 2020). In the 1970s, the WSP was present as early as July and persistent for three winters (Gordon, 1978; Carsey, 1980). In contrast, in 2017 the WSP was at its greatest spatial extent at the onset of austral spring, between

September and December. Recent analysis of sea ice thickness data has shown 'near polynya' events in 2010, 2013, 2014 and 2018 (Simmonds et al., 2021). During these events, sea ice thickness was reduced, but the ice concentration remained at standard levels.

Large WSP events like that of the 1970s are likely to have been rare in the past, as suggested by ice records, perhaps having only occurred once per century, although reconstructions are very uncertain (Goosse et al., 2021). Climate model projections suggest that in the future, with increasing atmospheric greenhouse gasses, the occurrence of the WSP will be even less frequent, due to an intensification of the haline stratification within the WSP region (de Lavergne et al., 2014). Studies suggest that the WSP may occur periodically, relating to the periodicity of long-term natural variability, such as the SAM index (Diao et al. 2022), and deep

ocean convection associated with the Atlantic Meridional Overturning Circulation (AMOC) (Martin et al., 2013; Jüling et al., 2018).

Many mechanisms have been suggested to trigger the onset of the WSP: deep convection of the ocean (e.g. Martinson et al., 1991, Martin et al., 2013), and upwelling at the Maud Rise (e.g. Cheon and Gordon, 2019; Rheinlænder et al., 2021), increased cyclone activity and wind stress (Francis et al., 2019; Campbell et al., 2019), the influence of atmospheric rivers (Francis et al., 2020),

katabatic winds from the continent (Smith et al., 2010) and a negative phase of the Southern Annular Mode (SAM) (Gordon et al., 2007). Despite limited observations, current understanding suggests that both ocean and atmospheric processes act in combination to form the precise preconditioning and trigger the appearance of the polynya (McHedlishvili et al., 2022; Cheon et al., 2014).

Once opened, the WSP permits an intense ocean-to-air heat flux in the cooler months, with the potential to influence atmospheric dynamics. However, the direct response of the atmosphere to the polynya has not been explored since the studies of Dare and

Atkinson, (1999, 2000) and Timmermann et al., (1999). Dare and Atkinson, (1999, 2000) investigated the response of the boundary layer (BL) and lower atmosphere to the polynya using an atmosphere-only model with prescribed Sea Surface Temperature (SST) and sea ice concentration (SIC). They notably showed that the heat released into the atmosphere by the polynya could enhance the turbulent mixing in the BL, increase the downward flux of momentum, and hence result in stronger surface winds over the polynya. This in turn would generate a pattern of divergence and downdrafts upstream of the polynya and convergence and updrafts

downstream of the polynya. It is important to note that Dare and Atkinson used a 2-dimensional model spanning 1000 m in height and 100 km across the polynya. Timmermann et al., (1999) hypothesized a low-level warming in response to the polynya and inferred the atmospheric circulation response from thermal wind balance assuming a level of no-motion at 6000 m height.

One may investigate the atmospheric response to the polynya from coupled climate models and atmospheric re-analysis. However, the interpretation of the results is difficult in such datasets as the atmospheric signal is a combination of the direct atmospheric

response and potential feedbacks.

Using the NCEP-NCAR reanalysis, Moore et al., (2002) estimated, for the 1970s WSP, an anomalous increase of up to 20 °C in air temperature above the WSP, approximately 20% more cloud cover, a reduction of sea level pressure by 6-8 hPa, an increase in precipitation of about 1 mm day$^{-1}$ and an increase in sensible and latent heat fluxes into the atmosphere of 150 Wm$^{-2}$ and 50 Wm$^{-2}$, respectively. For the smaller 2016/7 WSPs, Zhou et al., (2022) estimated an anomalous ocean-to-atmosphere heat flux of

approximately 40 Wm$^{-2}$. As these studies are based on a limited number of polynya events, the extracted signals could be significantly influenced by the weather variability. In addition, the re-analysis assimilates observations of the real atmosphere and hence potential secondary feedbacks; they cannot be interpreted as the direct response of the atmosphere to the polynya.

A composite analysis over three polynya events from the Community Earth System Model (CESM) demonstrate that the WSP in this model is associated with anomalies in the local turbulent heat flux, precipitation, and cloud formation, which significantly

impact the local radiative heat balance of the region (Weijer et al., 2017). However, without further information, the atmospheric

signal extracted by Weijer et al., (2017) cannot be interpreted as the direct response of the atmosphere to the polynya, but as a superimposition of this direct response and coupled feedbacks. In fact, Diao et al., (2022), who used the same model as Weijer et al., argued that the polynya is the result of a large-scale coupled ocean-atmosphere-sea ice mechanism (see also Kaufman et al., 2020). Coupled model studies suggest that the impact of the WSP and regional ice loss may be seen locally within the cyclonic region of the Weddell Sea through a moistening of the atmosphere and an enhanced low pressure (Diao et al., 2022), as well as further afield by influencing the Interdecadal Pacific Oscillation (IPO) via atmospheric signals reaching the western equatorial Pacific (Chang et al., 2020). Such coupled models often produce polynyas much larger than the largest observed polynya in 1974, which could result in an overestimation of their impacts on the atmosphere. In addition, it remains unclear to which extent the local response is influenced by ocean/ice feedbacks (such as the wind feedback on sea ice advection explored by Timmermann et al., (1999)) and to which extent the remote signal is a direct response to the polynya or is indirectly triggered by ocean processes (e.g. SST anomalies, changes in the overturning circulation).

Modelling studies investigating the atmospheric response to surface boundary perturbations or external forcing perturbations (e.g. $CO_2$, volcano) may be dependent on factors such as internal variability, varying physics schemes and model resolution, highlighting the importance of using multiple models and resolutions (e.g., Klaver et al., 2020). Atmosphere-only intercomparison projects reveal large differences between model responses to sea ice forcings, whereby in some cases the sign of the response is not robust (e.g., Ayres and Screen, 2019). Several reasons have been proposed on this, including but not limited to the position of the eddy driven jet (Bracegirdle et al., 2018; Holmes et al., 2019). Additionally, biases of the background phase of the SAM may lead to biases in the jet (e.g., Kidston and Gerber, 2010; Barnes et al., 2010). However, whether resolution makes a significant impact on a response in an atmosphere-only model is still not well understood. Streffing et al., (2021) demonstrate that when the same model at varying resolutions is forced with the same sea ice boundary conditions in polar regions, the higher resolution does not alter the results, even with more than 100 ensemble members.

In summary:

1) Current insights into the direct atmospheric response are limited to the local, 2d or geostrophic, response (Dare and Atkinson, 1999, 2000; Timmermann et al., 1999). The global 3-dimensional response of the atmosphere to the polynya using the full primitive equation dynamics remains to be determined.

2) Interpreting the direct response of the WSP in coupled models is made difficult by the potential impact of ocean-atmosphere-sea ice feedbacks. This is often compounded by oversized modelled polynyas, which likely overestimate the polynya's impact and hence feedback loops.

3) There are potentially large structural uncertainties in determining the atmospheric response to the polynya associated with model formulations (e.g., physics schemes, resolution, etc).

The aim of the present study is to address these issues. We emphasize that we are not concerned here by the preconditioning or the formation of the polynya. Rather, we seek to establish robust features of the global atmospheric response to the WSP once it is formed.

Following the approach of Dare and Atkinson, we employ atmosphere-only simulations with prescribed surface boundary conditions, but here with Atmospheric General Circulation Models (AGCM). By disallowing the ocean and sea-ice feedbacks, we can determine the one-way impact of the polynya on the atmosphere, which will help interpret atmospheric signals found in coupled simulations. To the best of our knowledge, this is the first study to determine the global 3d response of the atmosphere to the WSP. To establish the robustness of our results (e.g. structural uncertainties), we use three atmospheric General Circulation Models

(AGCM) and at two different resolutions for each model. Our experiments are conducted with the UK Met Office climate model, HadGEM3, the ECHAM5 Max Planck Institute model and the OpenIFS ECMWF model at high- and low- resolutions. We prescribe SIC and SST boundary conditions from the 1974-1975 polynya-year (July to June), i.e. a realistic sized polynya. The atmospheric response to the polynya is evaluated by comparing simulations with and without the WSP. Finally, updating on Moore et al. (2005), we compute the atmospheric signal associated with the 1974 WSP in the recent ERA5 reanalysis. This analysis is used to gauge the realism of our AGCMs outputs.

Section 2 describes the ERA-5 data, the three AGCMs used here and details of the simulations and methodology to extract the atmospheric response. Section 3 presents an analysis of our simulations while section 4 provides a discussion and implications for coupled studies. Conclusion are given in section 5.

## 2 Methods and Data

### 2.1 Data

The ECMWF Reanalysis version 5 (ERA5) monthly reanalysis (Hersbach et al., 2020) was chosen as the primary data source for the SIC and SST boundary conditions. In ERA5, the quality of 1974 data is slightly worse than data after 1979 as a consequence of the limited satellite observations at the time. ERA5 is a high-quality reanalysis product, with a 31km horizontal resolution. It has the limitations of relying on a model, rather than being observations only. However, for the purpose of this study, ERA5 is the best option to force the models with a realistic WSP forcing, due to the lack of reliable satellite products before 1979. The lower resolution of ERA5 is a limitation, where if using daily data, there would likely be inaccuracies around the edge of the polynya. Therefore, when choosing to use monthly or daily data, monthly data was thought to have been of higher quality. Crucially, for this study, high accuracy of SIC and SST is not critical as we compare twin experiments (with/without the WSP) and we do not expect that change of these boundary conditions in grid points bordering the WSP would significantly affect the atmospheric response.

Data from July 1974 to June 1975 are used to account for the WSP's formation starting in June until its dissipation in December (Fig. 1), then regridded via bilinear interpolation to the low- and high- resolutions grids of each model (Table 1). For comparison to a non-polynya-year, the same boundary conditions are used, but masked over the region of the polynya with non-polynya-year average (1982-2015) SIC and SST from May to December. The response to the WSP is defined as the difference between the two sets of simulations. The resulting SIC forcing is 100 % loss over the WSP in September and October, and up to 1 °C SST increase (Sup. 1). The SST under the sea ice is irrelevant for our setups, due to the absence of dynamic coupling.

### 2.2 Models

The WSP experiments are done with three different models: HadGEM3, ECHAM5, and OpenIFS, at their highest and lowest global horizontal resolutions.

We use the UK Met Office HadGEM3 high-resolution (N512) global atmosphere-only model configuration, submitted as part of phase 6 of the Coupled Model Intercomparison Project (CMIP6). HadGEM3 uses the GA7.1 (Walters et al., 2017) global atmosphere configuration, with the JULES GL7.0 land surface component model (Walters et al. 2017). The high-resolution N512 version of the model, hereafter HadGEM3-H, has 85 vertical levels and horizontal resolution of ~25 km at mid-latitudes and a

timestep of 15 minutes. The low-resolution N96 version of the model, hereafter HadGEM3-L, has a horizontal resolution of ~135 km, 85 vertical layers, and a time step of 20 minutes.

The experiments with the ECHAM5 model (Roeckner et al., 2003) are run using the high-resolution T255 version (horizontal resolution of ~50km, 62 vertical layers with a 100 second time step), hereafter ECHAM5-H, and the low-resolution T42 version (horizontal resolution of 300 km, 19 vertical layers with a 20-minute time step), hereafter ECHAM5-L. The resolution in both versions of ECHAM5 used here are comparably smaller than those used in HadGEM3 (see Table 1).

For both HadGEM3 and ECHAM5 models, the low-resolution models are integrated for 100 years with the repeat boundary forcing of the SST and SIC. For high-resolution, 15 and 30 years are simulated for the HadGEM3 and ECHAM5, respectively, for each boundary condition, due to their high computational costs at these resolutions. Atmospheric initial conditions were started arbitrarily and the number of repeating years with the identical SST and SIC forcing thus represents the number of ensembles.

The experiments with the OpenIFS are run using the high-resolution Tco399 version of the model, hereafter OIFS-H, with a horizontal resolution of ~25 km, 91 vertical layers, and 15-minute time step, while the low-resolution Tco95 version of the model, hereafter OIFS-L, is run at a horizontal resolution of ~100 km, 91 vertical layers, and 30-minute time step. The configurations are identical to OpenIFS-HRA and LRA in Savita et al., (2024) respectively. Both models are run as ensemble members, as opposed to transient runs used in HadGEM3 and ECHAM5 experiments. Each simulation starts 1 May 1988 and ends 30 Nov 1988. Each ensemble member is perturbed by 0.1 K noise in the initial skin temperature. The ensemble spread reaches a maximum after ~2 weeks, so that the members are clearly separated from 1 June onward. The differences in methods between the models are not expected to impact the results in any meaningful way. The atmospheric composition and solar forcing are as they were for 1988 (CMIP6 prescribed), but ozone and aerosol forcings are all from monthly climatology from CAMS.

## 2.3 Diagnostics and statistics

The response to the WSP is analysed as the transient/ensemble mean difference between the polynya and non-polynya simulations, disregarding the first two years for the transient experiments, and the first two months of the ensemble experiments. The ensemble average limits the influence of short-term weather events on our results and isolate the effect on the polynya on the atmosphere. Hereafter, the "response" will be used to refer to the ensemble-mean difference between simulations with and without the polynya. A students t-test is used to calculate statistical significance of the response and is reported at the 95 % confidence level. The Benjamini-Hochberg procedure (Benjamini-Hochberg, 1995) is then applied to the t-test p-values, to use a multiple-hypothesis test on all spatially correlated data and removing the assumption that point-by-point the results are significantly independent. The resulting significance is displayed as stippling.

Additional analysis is done to determine the responses dependence on the large-scale internal variability of the SAM index and the midlatitude tropospheric eddy-driven jet strength and position, all between 30 °W and 30 °E. The jet position and strength are calculated following the methodology of Ayres and Screen, (2019). This method uses maximum zonal wind values between 60 °S and 40 °S, between 30 °E and 30 °W, where the latitude of the maximum zonal wind is the position of the jet, and the maximum wind is the jet strength. Values outside of 3σ standard deviation is removed in calculating the correlations of the aforementioned large-scale internal variability to the responses to remove large outliers.

**2.4 Model result validation using ERA5**

We assess our results against the observation-based ERA5 reanalysis. To do this, we compute the difference between the 1974 (polynya-year) data and the mean 1980-2015 (non-polynya-years) data. This is of course a very limited sample, with just one polynya-year and is subject to high variability, and thus, we utilise the full ten ensemble member of ERA5. ERA5 assimilates observations which could include ocean-atmosphere-sea ice feedback processes, if they exist. One should be careful with comparing with our AGCM results which isolate the WSP impact on the atmosphere. Nonetheless, the ERA5 anomalies allow us to test some aspects of the modelled responses and establish whether they sit in a realistic dynamical regime. A normal distribution was calculated on the non-polynya-years and compared to the polynya-year, to determine the probability of the results occurring without the presence of the WSP.

**3 Results**

We start by analysing the low-resolution results, then explore and compare the high-resolution results, highlighting interesting similarities and differences. We then explore how these results correlate with the internal variability of the westerly jet and compare the results to ERA5 reanalysis for the same years used in the models.

**3.1 Spatial Component of the response**

The HadGEM3-L low-resolution spatial patterns of turbulent (sensible plus latent) heat flux response to the WSP (Fig. 2a-d) mimics that of SST and sea ice loss directly over the WSP, with an increase (upward flux) of up to 147 W m$^{-2}$ in August, followed by similar values in September. Despite the WSP not being yet at full spatial maximum (Fig. 1), the air-sea temperature gradient (not shown) is greatest during austral winter, hence the gradual decrease in turbulent heat flux from August to November. All responses are confined within the Weddell Sea region, and often to the WSP itself. In comparison with HadGEM3-L, the lower resolution ECHAM5-L turbulent heat flux response (Fig. 2e-h) shows a similar spatial pattern, peaking in September but at a lower maximum flux of 102 W m$^{-2}$. The OIFS -L turbulent heat flux response to the WSP (Fig. 2i-l) is again a comparable magnitude and spatial shape to HadGEM3-L with a maximum of 131 W m$^{-2}$, but peaking in September, analogous to ECHAM5-L.

The WSP turbulent heat flux response is highest between September and October, therefore, in the remaining of this study, we focus on the September-October mean response to the WSP for the sake of brevity. Monthly mean responses are noisier while not revealing more interesting features than the 2-month averages. For all three models, the extent of near-surface air temperature (TAS) response (Figure 3a-c) to the WSP is akin to that of turbulent heat flux response, with warming directly over the WSP region. There is in addition some warming over the surrounding Weddell Sea, notably to the north and east, which is most likely due to advection. For HadGEM3-L (Fig. 3a) the maximum temperature response reaches 7 K. The temperature response in ECHAM5-L (Fig. 3b), like heat flux, peaks at a smaller value of 4.5 K. ECHAM5-L also has the smallest spatial reach of the three. The near surface temperature is largest in OIFS-L (Fig. 3c), with a maximum of 7.5 K. In all three models, the temperature increase surrounding the WSP is limited to 2 K.

The mean sea level pressure (MSLP) response to the WSP in HadGEM3-L (Fig. 3d), shows a non-significant decrease directly over the WSP, with maximum of approximately -1.8 hPa, related directly to the surface warming. Away from the WSP, there is a smaller reduction in MSLP. The MSLP response in ECHAM5-L (Fig. 3e), is like that in HadGEM3-L over the WSP region, but with a smaller magnitude of -1.4 hPa. In OIFS-L (Fig. 3f), there is a decrease of ~0.7 hPa directly over the WSP, smaller than the

other two models, amongst a larger-scale positive anomaly in the region. We speculate, in part based on the lack of statistical significance, that the large-scale positive anomaly is due to internal variability and partially cancels the localised MSLP directly over the WSP. An analogous (but opposite) effect occurs in ECHAM5, overestimating slightly its peak MSLP response over the WSP.

Maximum precipitation significantly increases by up to 0.7 mm day$^{-1}$ over the WSP region for HadGEM3-L (Fig. 3g). The September – October average total sum of precipitation directly above the WSP is ~3. 7 mm day$^{-1}$. This response could be explained by the increased surface latent heat flux, as the evaporated water content correspond to ~4 mm day$^{-1}$ if precipitated back over the WSP. The ECHAM5-L precipitation response (Fig. 3h) is weaker at a maximum of 0.5 mm day$^{-1}$ and a total of ~2 mm day$^{-1}$ directly above the WSP. The spatial extent of the significant response is also more localized directly above the WSP. The OIFS-L precipitation response (Fig. 3i) is of a higher magnitude up to 0.9 mm day$^{-1}$ and totals to ~20 mm day$^{-1}$ directly above the WSP, which is significantly more than the other models, due to a consistently higher value over the whole of the WSP. Savita et al. (2023), show that precipitation is overestimated in the OIFS models, which may explain the larger response. There is a small significant reduction of 0.1 mm day$^{-1}$ in precipitation in the area north of the WSP.

The geopotential height at 850 hPa (Z850) shows no significance and lacks consistency between models. In HadGEM3 (Fig. 3j), there is an increase of 7 m. For ECHAM5 (Fig. 3k), the response is almost negligible, with a slight decrease over the WSP. The OIFS-L response (Fig. 3l) shows a clearer increase in geopotential height over the WSP of 8.6 m.

As expected, the zonal wind response at 850 hPa (U850) is associated with the changes in geopotential height at the same level through geostrophic balance. In HadGEM3-L (Fig. 3m), there is a reduction in the westerly wind north of the WSP of -0.8 ms$^{-1}$ and an increase in the south of 0.6 ms$^{-1}$, consistent with an anticyclonic response to an increase in geopotential height (Fig. 3j). The response of the ECHAM5-L zonal wind at 850 hPa (Fig. 3n), shows almost the opposite, cyclonic, response to HadGEM3-L, however this is small and not significant. The response in OIFS-L (Fig. 3o) is similar to HadGEM3-L, although not significant, but is consistent with the geopotential height again (Fig. 3i). While there is a significant reduction in the westerly winds to the north of -0.6 ms$^{-1}$, there is a smaller insignificant increase in zonal wind to the south of 0.5 ms$^{-1}$. In summary, the response of the wind and geopotential height aloft are less robust and less significant than the direct surface processes.

To explore the vertical structure of the response in more details, we compute the 10 °W to 10 °E (directly over the WSP) zonal-mean potential temperature (TA) response for the September-October means for HadGEM3-L, ECHAM5-L and OIFS-L (Fig. 4a-c). Both HadGEM3-L and OIFS-L exhibit a warming of up to 3 K in the lower troposphere below 900 hPa, with a smaller increase (~1 K) aloft (not significant), up to 700 hPa (or ~1.5 km). ECHAM5-L show a similar vertical structure but only reaches half of the magnitude of the two other models with a maximum temperature response of 1.5 K near the surface.

The HadGEM3-L and OIFS-L response in zonal-mean geopotential height (ZG) over the WSP (Fig. 4d&f, respectively) exhibit a similar increase of about 20 m to the South of the WSP, significant throughout the troposphere, and a smaller decrease to the north. ECHAM5-L (Fig. 4e), however, shows a smaller increase to the north up to 350 hPa, and a small surface increase to the south of the WSP. The zonal mean geopotential height response shows a barotropic shift above the WSP, however this is small in magnitude and inconsistent across models.

Throughout the troposphere, the zonal-mean zonal wind (UA) again shows a consistent geostrophic response with geopotential height (Fig. 4g-i). HadGEM3-L and OIFS-L show a decrease in the winds north of ~65° S, although HadGEM3-L is more localised to the WSP, with another sign change north of 50 ° S. However, ECHAM5-L shows an increase in zonal velocity from 65°S to 55°S flanked by negative anomalies. These differences in the zonal wind and geopotential response may be associated with the

position of the tropospheric eddy driven jet relative to the WSP within each model, where there is a ~5° difference in HadGEM3. Nonetheless, these results further demonstrate the lack of robust response and limited significance to the WSP above 500 hPa, indicating no evident atmospheric pathway for the response away from the Weddell Sea region.

## 3.2 Comparison to high-resolution versions

For the high-resolution simulations, the magnitude in turbulent heat flux response (Fig. 5) is notably higher in all three models when compared to corresponding low-resolution versions. The biggest increase is found in HadGEM3-H with a difference of 45 Wm$^{-2}$ (Fig. 5a-d), then ECHAM5-H with 38 Wm$^{-2}$ (Fig. 5e-h), and OIFS-H with 33 Wm$^{-2}$ (Fig. 5i-l). Despite this increase, the high-resolution turbulent heat flux maintains the same spatial and temporal pattern as the low-resolution response, weakening into October and November.

Fig. 6 summarizes the response for all six model versions used for this study in the turbulent heat flux, near surface temperature, precipitation, MSLP, geopotential height and zonal wind at 850 hPa. The responses are averaged monthly in time and spatially over the WSP region (10° E – 10° W, 63° S – 68° S, see Fig. 1). All models agree on the sign of the response for all (near) surface variables but reveal ambiguous results aloft. For HadGEM3 and ECHAM5, the high-resolution versions of the model are notably greater than their low-resolution counter parts by up to 50 % while the OIFS-H response is larger than the OIFS-L response by only up to 30 %.

The sign of the MSLP response is almost consistently negative (Fig. 6d), as shown by the ensemble mean, however the signal to noise ratio is low. Geopotential height and zonal wind at 850 hPa are further exasperated, as the response time series fall on both sides of zero. There is no longer a majority agreement on the sign, with the ensemble mean fluctuating above and below zero, confirming the lack of a robust response to the polynya in the free troposphere.

## 3.3 Jet and correlations

As discussed in the introduction, we expect that the systematic biases as well as the natural variability of the models may influence the response to the polynya. Specifically, a stronger or more poleward jet (or a more positive SAM index) would increase the latent heat flux response directly above the WSP, and thus, the magnitude of the overall atmospheric response.

The mean state of the jets differs significantly across the models. Of the low-resolution models, HadGEM3-L has the most southward jet averaging at 51.3 °S, and the most northward jet is found in ECHAM5-L averaging at 46.3 °S. OIFS-L has the fastest jet of 14.5 ms$^{-1}$ (Sup. 2a&b). The averaged SAM indices are the most positive in the high-resolution models, with averages ranging between 0.2 for ECHAM5-H and 0.4 for HadGEM3-H, compared to consistent averages of 0.1 across the low-resolution models (Sup. 2c).

We now regress the atmospheric response to WSP in the ensemble members against the mean jet strength, position, and SAM index in the polynya experiment (Fig 7). The turbulent heat flux response shows a weak correlation (Fig. 7) with the SAM index in ECHAM5-L (R2 = 0.11, P>0.05) and OIFS-L (R2=0.06, P<=0.05) but no relationship is found in HadGEM3-L (R2 = 0.00, P>0.05) (Fig. 7a-c). However, there is a weak correlation with the jet strength for HadGEM3-L (R2=0.04, P>0.05). The response in MSLP (Fig. 7d-f), geopotential height at 850 hPa (Fig. 7g-i) and zonal wind at 850 hPa (Fig. 7j-l) shows a higher correlation with the SAM index (albeit not robust across models) than the jet strength and latitude. For all variables but zonal wind, there are

no significant correlations of the atmospheric response to WSP and jet position. We note that the zonal wind response to WSP has a significant correlation to jet position but not jet latitude in ECHAM-L ($R2=0.06$) and OpenIFS-L ($R2=0.12$).

These results indicate that the variability between the resolutions of the same model is dictated by the SAM index variability (Sup 2b), however, the variability between the different models is further dictated by a combination of the strength and position of the jet. The high-resolution results were not included here, due to the small ensemble numbers, leading to a limited comparison (Sup 290    3).

The overall magnitude of the response shows some correlation to the mean state of the SAM index and the strength of the tropospheric eddy driven jet, and to a lesser extent, the position. Although, where ECHAM5 has the most northward jet, the overall response to the WSP was lower and there is more dependence on the position of the jet latitude. Additionally, ECHAM5 has a near 50 % increase in magnitude in the high-resolution version, whereby the jet latitude is on average 4.3 ° more southward. Jet position 295    and jet strength are not linearly related. ECHAM5 has an equator ward bias in the position of the jet (Barnes et al., 2010; Kidston and Gerber, 2010), associated with persistence of SAM, whereas OpenIFS and HadGEM3 have a lesser bias.

### 3.4 Comparison to ERA5 reanalysis

To evaluate the models' reliability, we estimate the observed atmospheric anomalies in 1974, the same year as that used for the SIC and SST boundary conditions in our simulations using the ERA5 reanalysis (see section 2.4 for details). ERA5 is a reanalysis, 300    and may be subject to model biases, especially before the extensive assimilations of satellite observations from 1979 onwards. Additionally, precipitation and heat flux are not assimilated, where outputs are highly dependent on the bulk formula used. We use all ten ERA5 ensemble members. Nonetheless, the 1974 turbulent heat flux, surface temperatures and precipitation anomalies over the WSP in ERA5 (Fig. 8a) are similar to the model responses spatially, and, of a similar magnitude. Averaged over the WSP region (red box, Fig. 1), the heat flux, surface temperature and precipitation anomalies are about 70 Wm, -3 K, and 2 mm day$^{-1}$ 305    above average. The MSLP anomaly (Fig. 8d) shows a less obvious pattern compared to the other fields, but the imprint of the polynya as negative anomaly (of about -3 hPa) is noticeable.

To put these anomalies into context, histograms of the variables (averaged over the WSP, red box in Fig. 1) are constructed for the years 1980-2015 (no polynya years) and the 1974 states are highlighted (with the pink star; see Fig. 8, bottom row). The 1974 heat flux state sits well outside of the range of possible heat fluxes in non-polynya years. The 1974 surface temperature, MSLP, and 310    precipitation states are found at the extreme range of possible realizations in non-polynya years. Using a Gaussian fit, we estimate that all three 1974 variables have a less than 5% chance of occurring naturally in non-polynya years. Going further, a joint PDF of precipitation and MSLP shows that the 1974 state has a probability of 0.2 % of occurring outside of a polynya year (Sup. 4). These results are consistent with the NCEP-NCAR reanalysis results from Moore et al., (2002).

Overall, the low probability of occurrence of the 1974 near-surface state in non-polynya years suggests that it can be interpreted 315    as primarily a response to the polynya opening (although possibly including some ocean-sea ice feedbacks). This suggests that our AGCM results are within a 'realistic' threshold from the reanalysis data for the same years.

### 4 Discussion

In both the low- and high-resolution simulations, the presence of the WSP generates a turbulent heat flux from the ocean to the atmosphere between 100 and 190 Wm$^{-2}$, the magnitude being model dependent. Decreasing from September to November due to

the seasonal reduction of the ocean-atmosphere temperature gradient, the localised heat flux leads to increased atmospheric surface temperature up to 700 hPa and to increased precipitation. The increase in temperature, although not extending beyond the lower troposphere, is associated with a barotropic shift in geopotential height aloft, however this is not outside of the bounds of internal variability. The modelled response to the WSP is similar to, and complements that found in previous studies (Weijer et al., 2017), a study that assesses the response to the polynya in a coupled model. The models show strong agreement in the locality of the response, and all agree that there is a reduction in surface pressure over the WSP (albeit not significant), again agreeing with the early literature (Timmermann et al., 1999). Aloft in HadGEM3 and OpenIFS, this is reinforced by an increase in geopotential height at 850 hPa and a geostrophic anticyclonic response in the wind around the WSP, further agreeing with theories from the literature that the WSP can influence local dynamics (Dare & Atkinson, 1999, 2000).

The high-resolution versions of the models show a similar spatial pattern in all models, maintaining the horizontal and vertical locality of the responses seen in the low-resolution versions of the experiments. However, the magnitude of the response is up to 50 % higher in the high-resolution versions of the models. OpenIFS is the least resolution-sensitive model, with response to WSP only increasing by a maximum of 30 % between the two resolutions. HadGEM3 shows the greatest differences between resolutions, but HadGEM3-H is the most subject to internal variability due to the smaller ensemble and is therefore less robust than the other models. ECHAM5 also shows large differences between the two resolutions, this may be due to the models having the biggest differences, i.e., time step and vertical resolution.

However, the response to the WSP is not entirely robust across the three models. In particular, the responses above the boundary layer are weak, often not significant, and inconsistent across models and resolutions. For example, ECHAM5 exhibits a nearly opposite response aloft to the two other models at low resolution. We speculate that the responses aloft are in fact traces of the equivalent barotropic variability that dominates the natural monthly-to-yearly variability of the Southern Hemisphere (Thompson and Wallace, 1999). We expect that with an increasing number of members/realizations, the "response" identified in the free troposphere would converge to near zero across all models.

The jet is predicted to shift more poleward and increase in strength as greenhouse gases increase (Smith et al., 2017). Despite the prediction that future WSP events will be less frequent and smaller (De Lavergne et al., 2014), our jet analysis suggests that with a poleward shift and strengthening of the jet, a smaller WSP may still induce a sizable, yet local response in the lower troposphere.

The response in our models shares many similarities to the observational reanalysis of Moore et al., (2002) who used the NCEP-NCAR reanalysis. While assimilated observations for 1974 have not changed much since then, the quality of reanalysis products has significantly improved since the NCEP-NCAR release, and we expect that our repeat analysis on the recent ERA5 product provides more reliable results than those of Moore et al., (2002). Our analysis and that of Moore et al., (2002) agree on the sign and locality of the atmospheric signal associated with the WSP, however the magnitude differs. For example, the NCEP-NCAR reanalysis has approximately twice the surface air temperature compared to the ERA5 analysis (20 °C versus 10 °C). Notably, the anomalies inferred from ERA5 reanalysis are in closer agreement with our AGCMs results, specifically of the high-resolution models, where ERA5 has a similar horizontal resolution of ~30 km. However, the total turbulent heat flux response of our AGCM study is 25 % less than that the anomalies seen in ERA5, where sensible and latent heat were close to equal in our study, as seen in our precipitation response, which is 0.5 mm day$^{-1}$ larger than the observational reanalysis.

Subject to limitations and biases associated with the NCEP-NCAR and ERA5 reanalysis, and the use of only one polynya realization in 1974 to extract the atmospheric response, our AGCM responses are mostly in agreement with the reanalysis although aligning more closely with the weaker responses inferred from the recent ERA5 product. Our results suggest that the anomalies

seen in NCEP-NCAR and ERA5 reanalysis are primarily the direct response to the polynya, and, if any, coupled feedbacks make a small contribution. This comparison to reanalysis reinforces our conclusions that the direct response to the WSP is strong and significant but highly localized to the boundary layer just above the WSP. In addition, our results show that the response has little memory in the atmospheric-only system (i.e., local in time too) and vanishes rapidly with the polynya.

Our findings are in contradiction with some previous studies which suggest that the response to the WSP may have a remote reach as far as the tropics (e.g., Chang et al., 2020). By design, our experiments isolate the direct response to the WSP eliminating any potential feedbacks due to two-way interactions with ocean and sea ice. It is possible that further afield impacts to sea ice change and atmosphere may require such feedbacks with the ocean at lower latitudes (e.g., England et al., 2020; Ayres et al., 2022). We also show that there is no significant impact on the westerly mid-latitude jet, which may interact with the wider climate through teleconnections (e.g., Wang and Cai, 2013; Fogt et al., 2011). In our experiments, there is no clear pathway through the atmosphere alone for a remote response. Our response may be limited by our use of monthly model data, as opposed to daily, where daily SST fluctuations can impact the atmospheric response to ocean anomalies (Zhou et al., 2015).

Two recent studies (Kaufman et al., 2020, Diao et al., 2022), through the analysis of coupled climate simulations, have suggested that the WSP could be part of complex coupled ocean-atmosphere-sea ice modes of variability. In both cases, the modelled polynyas are much larger than the largest observed polynya found in 1974. While the atmospheric signals associated with the polynya in these coupled models are also largely confined above the polynya, by virtue of their sizes the modelled polynya drive large-scale atmospheric anomalies projecting more strongly on oceanic features. For example, Diao et al., (2022)'s coupled mechanism relies on a change of the wind-stress curl and precipitation patterns on the scale of the Weddell gyre. Our results show responses with significantly smaller spatial scales even for a 1974-size polynya. This suggests caution in analysing model polynya dynamics where oversized polynyas may generate unrealistic coupled feedbacks and over-estimate the role of coupled dynamics in maintaining the polynya recurrence.

In summary, the WSP may not interact, to first order through the atmosphere, with the climate on a large scale and may be too far south and within sea ice edge for coupled feedbacks to have a substantial impact on results. However, the phase of the SAM and jet may make the response bigger or smaller. Resolution may be less important than internal variability in modelling the atmospheric response to ice loss events in AGCMs. The study was limited by the ensemble size in the high-resolution model because of this. Nonetheless, the modelled responses to the WSP bear strong similarity with anomalies seen in atmospheric reanalyse data and previous studies.

## 5 Conclusions

Our study uses three AGCMs to determine the direct response of the atmospheric circulation to the 1974 WSP, using sea ice and SST boundary conditions from the ERA5 reanalysis. For each AGCM, we employ low- and high-resolution versions of the model to assess the dependence of the response to the WSP to resolution (ranging from 25 to 300 km).

We show that the AGCM response to the WSP is localized to the WSP region, vertically restrained to the boundary layer and is only present in the late austral winter and spring barely outlasting the WSP itself. The WSP creates up to 150 $Wm^{-2}$ turbulent heat flux from the ocean to the atmosphere, leading to a warming of the atmosphere at the surface of up to 10 K in August and September. A warming (of less than 3 K) spreads up to 10° northeast via advection. A small amount of associated increased surface moisture flux does advect away from the region, however, the majority of precipitation occurs directly over the WSP. While not statistically significant in each model, the dynamical response shows a robust low-level baroclinic structure with a small (~1 hPa) localised

decrease in surface pressure over the WSP, and an increased geopotential height at 850 hPa associated with a geostrophic cyclonic response in the winds. The response found in the free troposphere is relatively weak and inconsistent across models. We suspect that these responses are strongly influenced by the natural variability of the jet due to the small signal-to-noise ratio above the boundary layer.

The patterns of the responses at low levels show many similarities between the low- and high- resolutions versions. However, the responses in the high-resolution versions of the models appear to be of larger magnitude than in their low-resolution counterparts. The sensitivity of the response to horizontal resolution may be due, in part, to internal variability associated with the mean state of the SAM index and the position of the westerly mid-latitude jet as the high-resolution simulations had a low number of ensemble members. This is especially true when considering the response in the free troposphere where the response (if any) is much smaller than internal variability.

Overall, the responses we obtained are consistent with previous studies (e.g., Moore et al., 2002; Weijer et al., 2017; Thompson and Wallace, 2000; Timmermann et al., 1999; Dare and Atkinson, 2000, 1999), and with the anomalies found in ERA5 reanalysis for the same years used to force our AGCM experiments. It is worth noting however that the larger responses found in the high-resolution models are in better agreement with the ERA5 reanalysis.

By design (use of AGCMs with prescribed SST and sea ice boundary conditions), our study extracts the direct response to the WSP and shows that it is highly localized to the WSP. There is no clear atmospheric path to propagate the response away from the WPS region in our simulations. The direct response does not expand further north than the sea ice edge, thus, it is unlikely that the warming would reach the open ocean without a deep ocean pathway (e.g., Gordon 1978, Chang et al., 2020).

Nonetheless, our modelling framework does prohibit potential ocean/sea ice coupled feedbacks which may contribute to the propagation the response further afield. It seems unlikely however that such coupled feedbacks could strengthen much the atmospheric response (or in an unrealistic manner) since the magnitude of the response in our experiments are consistent with those inferred in previous studies and ERA5. We do warn however that inferences from unrealistically large polynya occurrence in coupled models should be taken with caution. Such polynyas would drive too wide an atmospheric response that could in turn overestimate ocean and sea ice feedbacks and the coupled nature of the polynya dynamics. For example, our results suggest that a

1974-size polynya (the largest observed) cannot generate a first order atmospheric response that would project on the scale of the Weddell gyre.

Further investigations with coupled models where, for example, feedbacks are turned on and off could help clarify the role of these feedbacks.

**Competing interests**

The contact author has declared that none of the authors have any competing interests.

**Author contributions**

Experiments were designed by HA, DF, WP, with input from JK and MO. Models were run by HA, WP and JK. Analysis and writing of the manuscript was done by HA, with feedback from all authors.

**Acknowledgments**

This research is funded by the European SO-CHIC project (no. 821001) funding from the European Union's Horizon 2020 research and innovation programme. For more information on SO-CHIC, please visit http://www.sochic-h2020.eu/. W. P. acknowledges support by GEOMAR and IBS (IBS-R028-D1). J. K. is supported by JPI Climate/Ocean (ROADMAP project grant 01LP2002C).

**Open Research**

Model output data can be accessed: Ayres, H., Park, W. and Kjellsson, J.: Data from polynya models for paper 'The atmospheric response to the Weddell Sea Polynya and its resolution dependence in three Atmospheric General Circulation Models'. University of Reading. Dataset. https://doi.org/10.17864/1947.000487, 2023 and Hersbach, H., et al.: ERA5 monthly averaged data on single levels from 1940 to present. Copernicus Climate Change Service (C3S) Climate Data Store (CDS), DOI: 10.24381/cds.f17050d7, 2023

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

**Table 1: Summary of model experiments. Each model was run with polynya-year and non-polynya-year boundary conditions.**

| Models | Resolution | Members / Years |
|---|---|---|
| *HadGEM3* | N96 (135 km) | 100 |
| | N512 (25 km) | 15 |
| *ECHAM5* | T42 (300 km) | 100 |
| | T255 (50 km) | 30 |
| *OpenIFS* | $T_{co}95$ (100 km) | 100 |
| | $T_{co}399$ (25 km) | 30 |

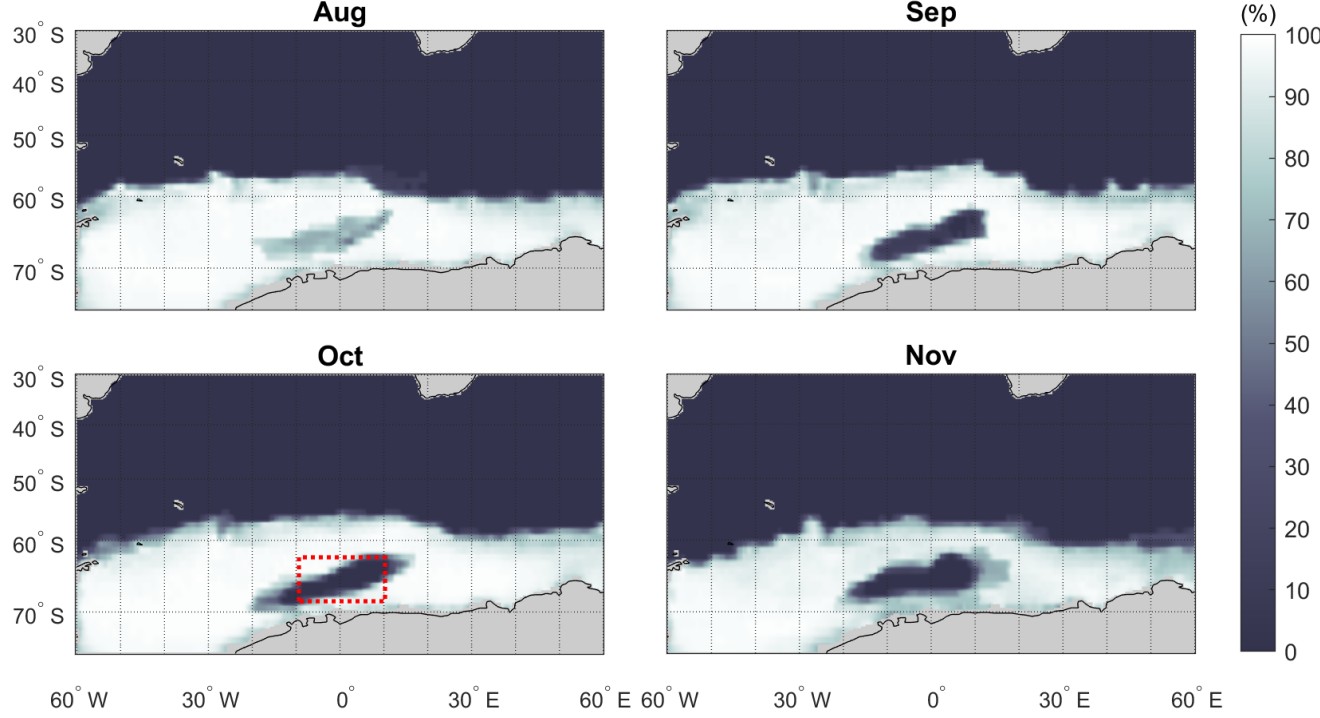

**Figure 1: Monthly sea ice concentration from ERA5 dataset for August to November 1974. The red box in the October panel highlights the 10° E – 10° W, 63° S – 68° S region, used for the mean responses in Figure 6 and onwards.**

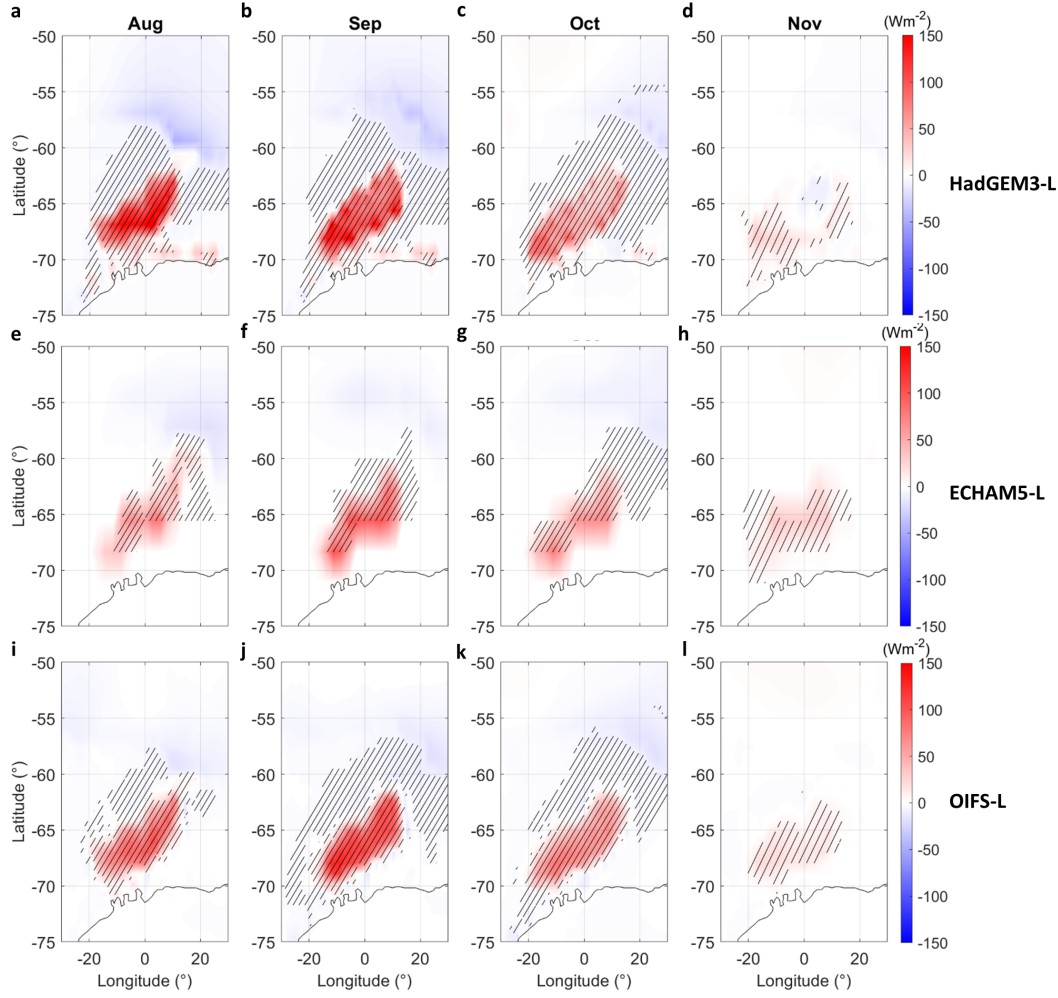

Figure 2: (a-d) The low-resolution turbulent heat flux (positive into the atmosphere) response to the WSP (Polynya simulation minus non-polynya simulation) for HadGEM3-L from August (a) to November (d). (e-h) as (a-d) but for ECHAM5-L. (i-l) as (a-d), but for OIFS-L. Turbulent heat flux is calculated as the combined sensible and latent heat flux. Stippling indicates the 95% significance level by t-test and the Benjamini-Hochberg procedure.

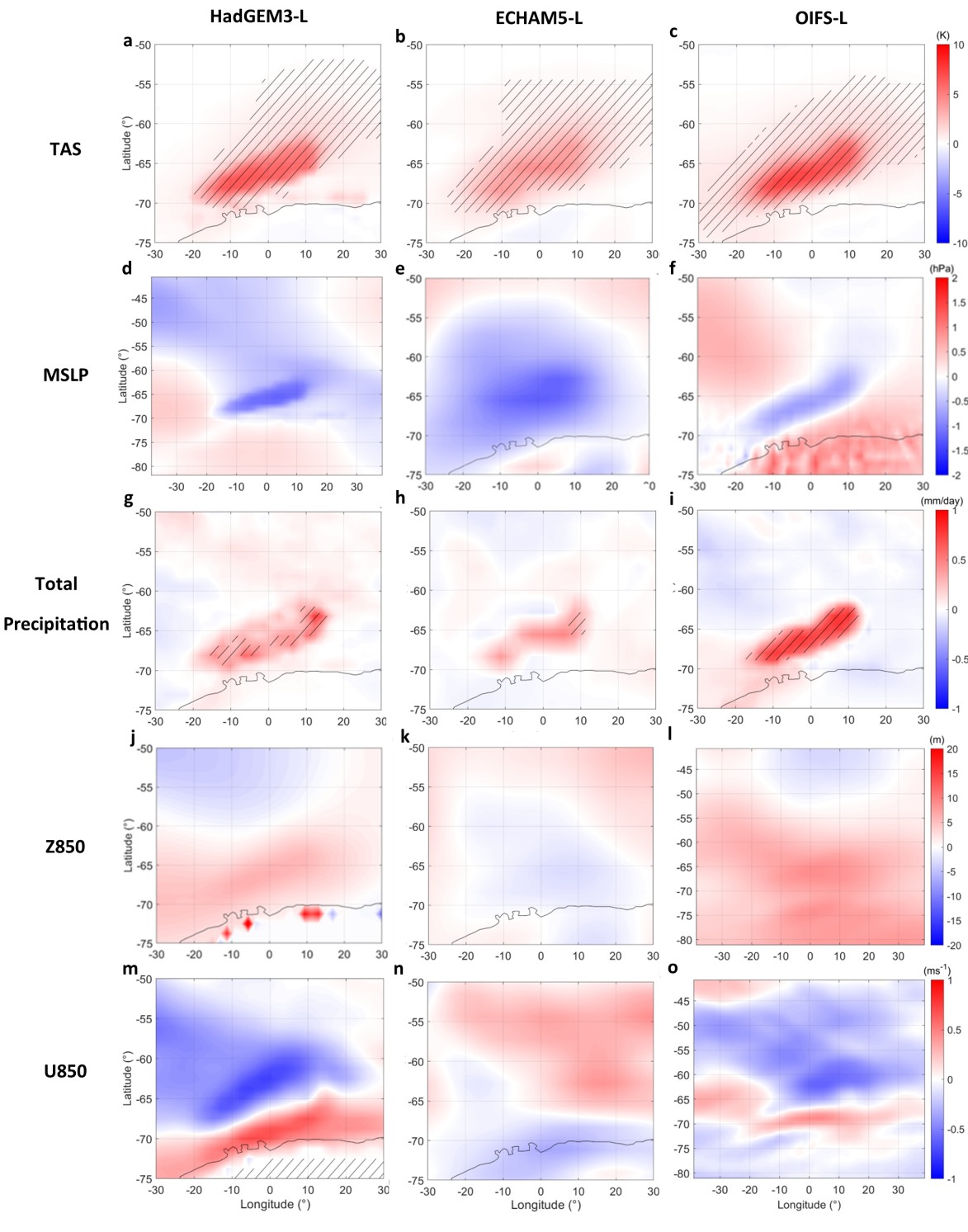

**Figure 3: Near surface temperature (TAS) September-October mean response for HadGEM3-L (a), ECHAM5-L (b), and OIFS-L (c). (d-f) as for (a-c) but for MSLP. (g-i) as for (a-c) but for total precipitation. (j- l) as for (a-c) but for geopotential height at 850 hPa (Z850). (m- o) as for (a-c) except for zonal wind at 850 hPa (U850). Stippling indicates the 95% significance level by t-test and the Benjamini-Hochberg procedure.**

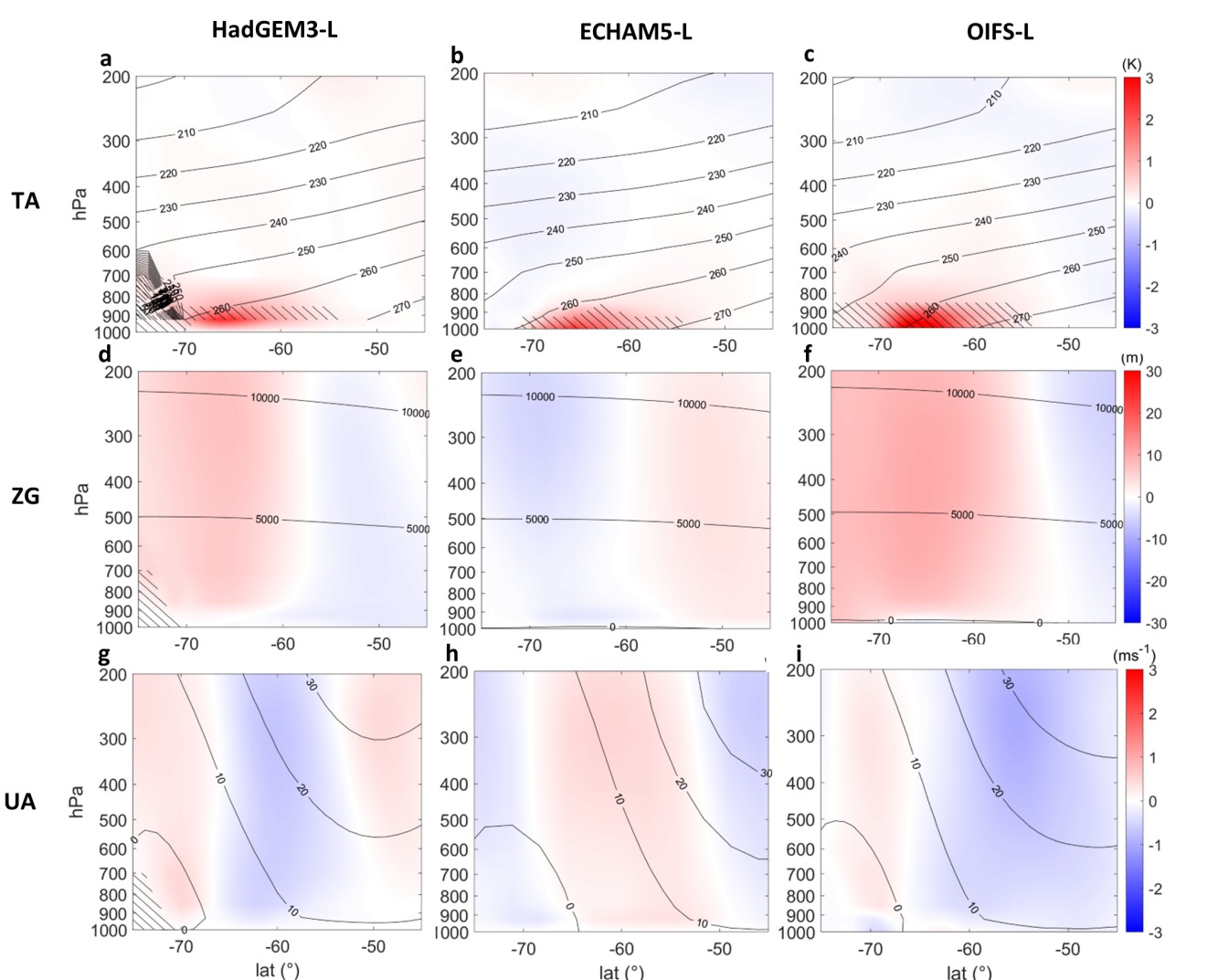

**Figure 4: Zonal-mean temperature (TA) over the WSP region (10 °W to 10 °E) September-October mean response for HadGEM3-L (a), ECHAM5-L (b), and OIFS-L (c). (d-f) as for (a-c) but for geopotential height (ZG). (g-i) as for (a-c) but for zonal wind (UA). Stippling indicates the 95% significance level by t-test and the Benjamini-Hochberg procedure..**

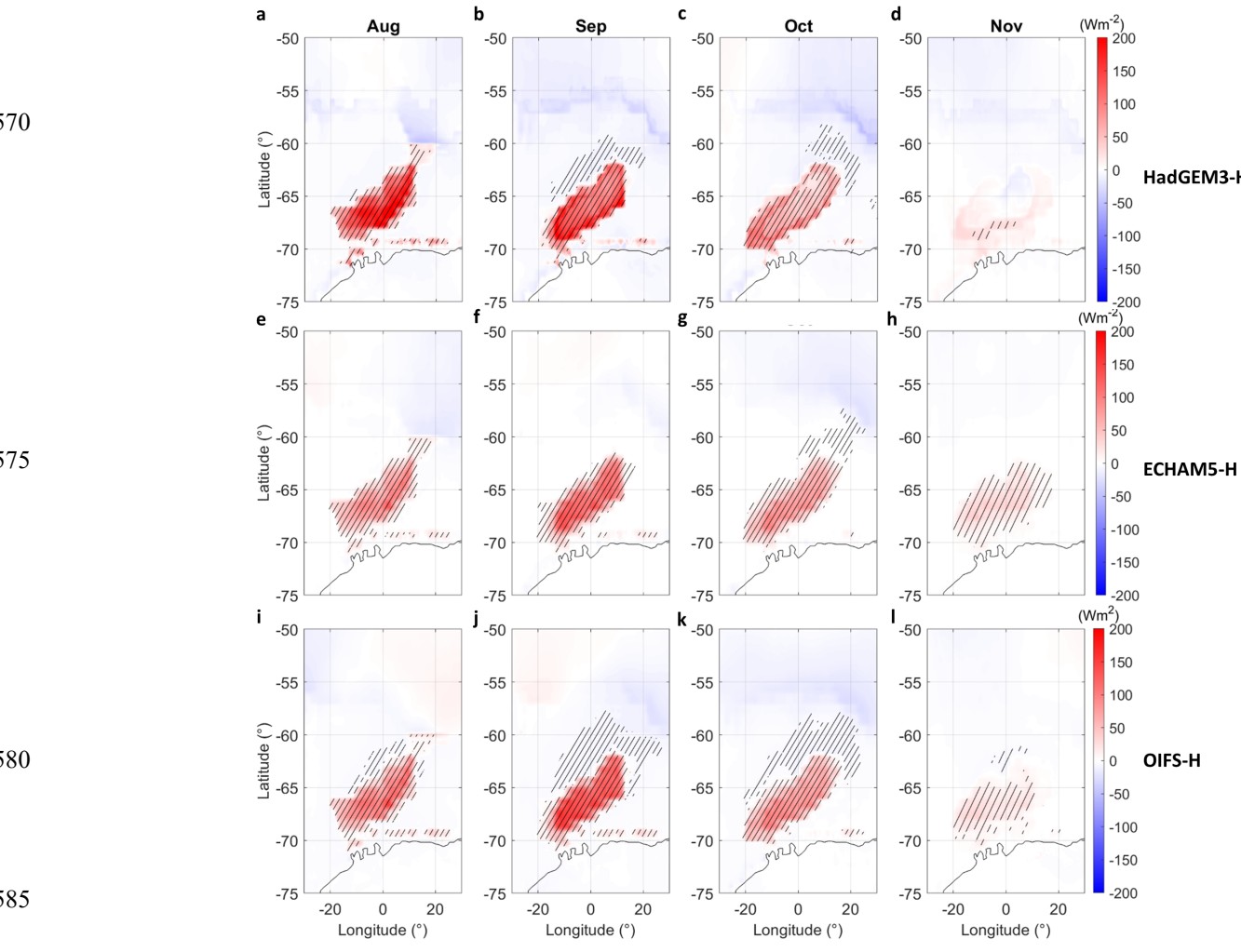

**Figure5: The high-resolution turbulent heat flux (positive in the atmosphere) response to the WSP for HadGEM3-H from August (a) to November (d). (e-h) as (a-d) but for ECHAM5-H. (i-l) as (a-d) but for OIFS-H. Turbulent heat flux is calculated as the combined sensible and latent heat flux. Stippling indicates the 95% significance level by t-test and the Benjamini-Hochberg procedure.**

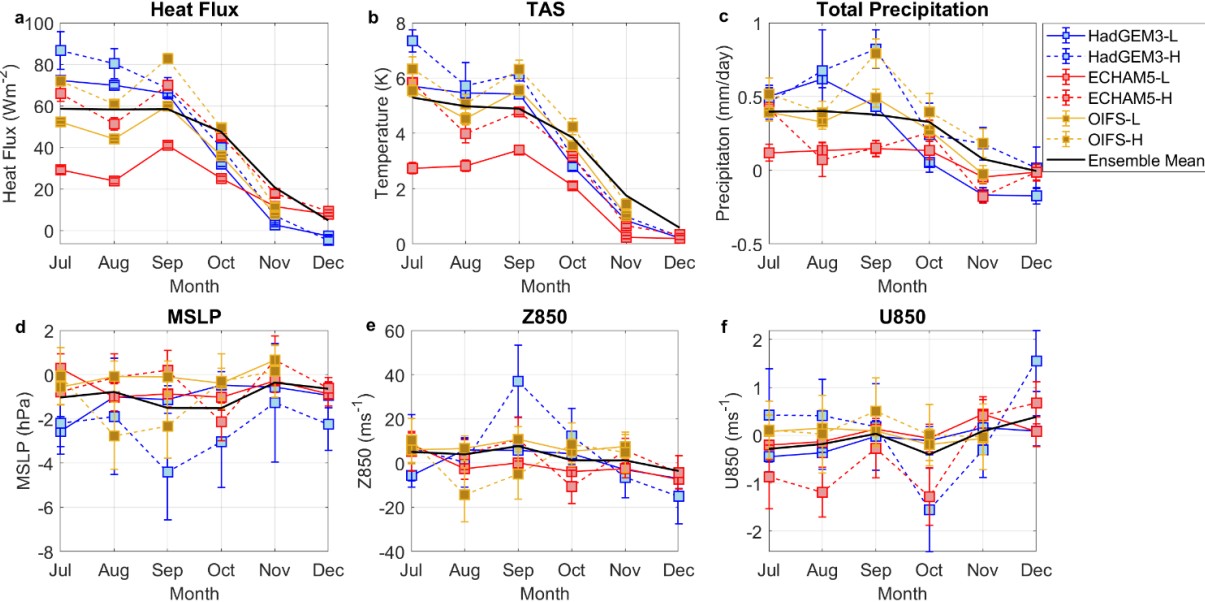

**Figure 6: (a) Monthly mean and standard error of heat flux response to the WSP averaged over (10° E – 10° W, 63° S – 68° S) (red box in Figure 1), with HadGEM3 (blue), ECHAM5 (red) and OIFS (yellow). Low-resolution and high-resolution are denoted by continuous and dashed lines, respectively. The ensemble mean of all three models and both resolutions is denoted by the black soldi line. (b) as (a) for surface temperature. (c) as but (a) for precipitation. (d) as (a) but for MSLP. (e) as (a) for geopotential at 850 hPa. (f) as (a) but for zonal wind at 850 hPa.**

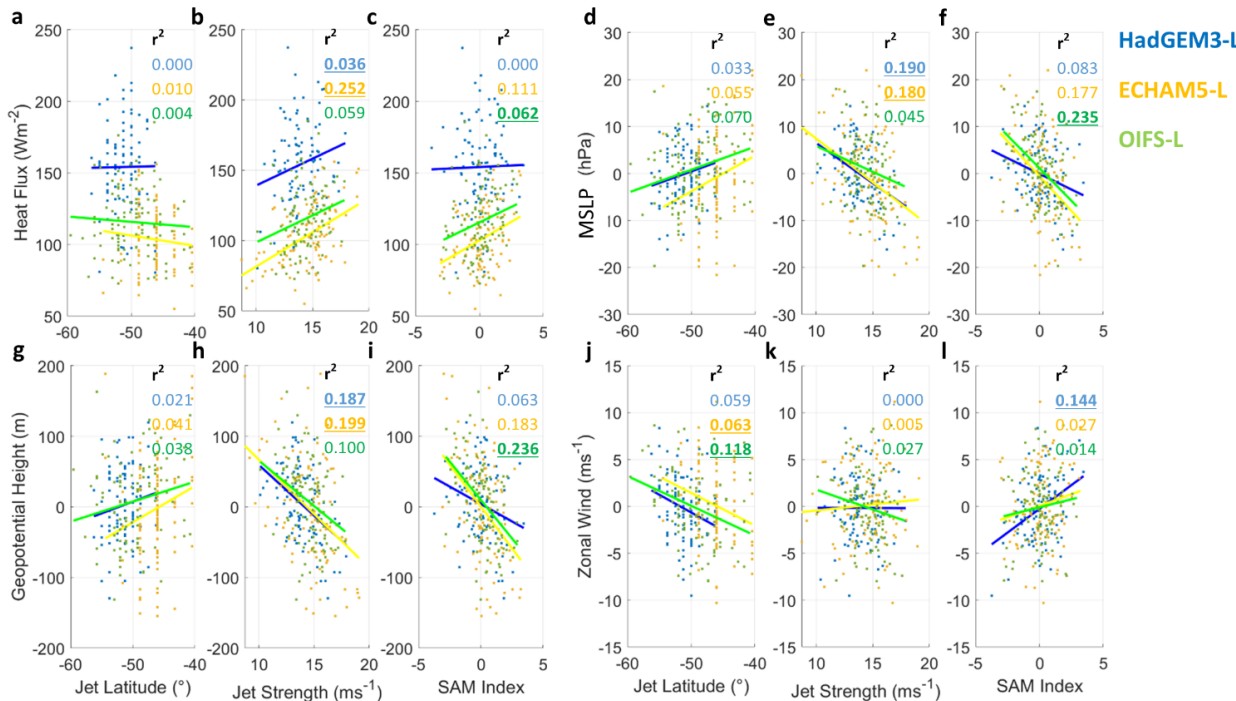

**Figure 7: Heat flux response to the WSP (averaged over 10° E – 10° W, 63° S – 68° S; red box in Fig. 1) against jet latitude (a) jet strength (b), and SAM index (c) for the low-resolution versions of the models. Blue for HadGEM3-L, yellow for ECHAM5-L and green for OIFS-L. Each dot represents each ensemble member for September. (d-f) as (a-c) but for MSLP, (g-I) as (a-c) but for geopotential height at 850 hPa, and (j-l) as (a-c) but for zonal wind at 850 hPa. Bold underline shows the dominant processes for each model.**

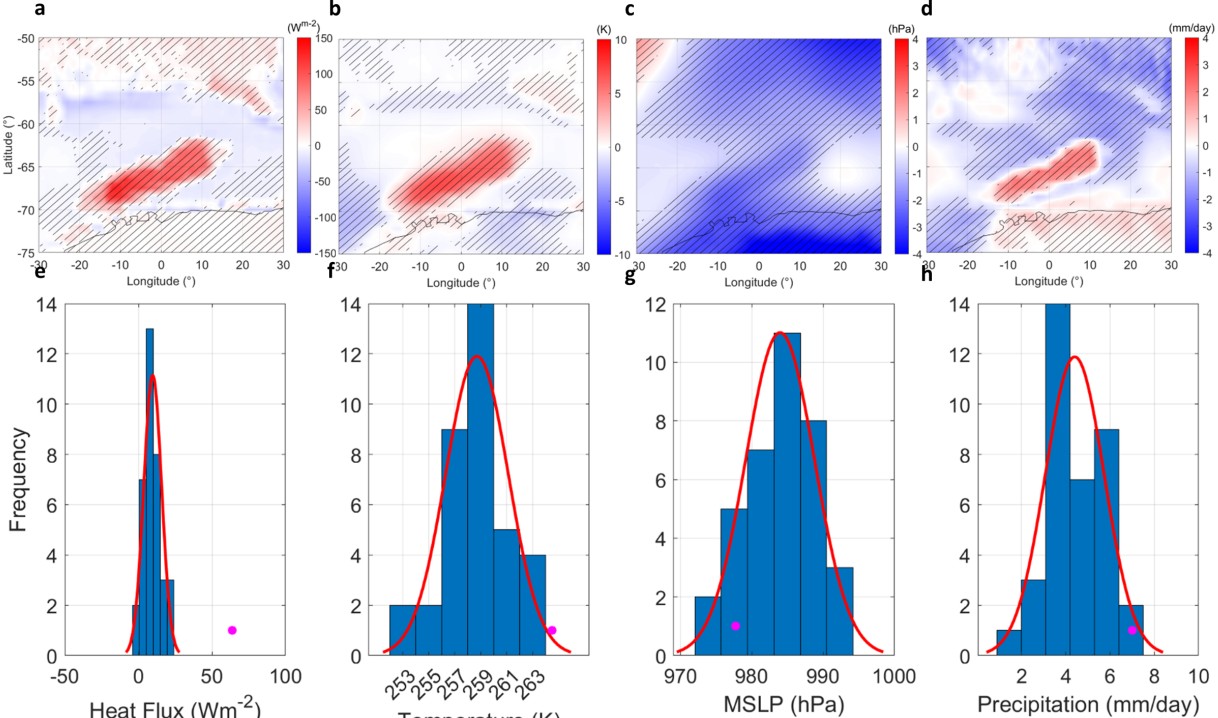

**Figure 8: ERA5 heat flux (a) difference between 1974 and the 1980-2015 average for September-October mean. (b) as (a) for surface temperature. (c) as (a) but for MSLP. (d) as (a) but for precipitation. Stippling indicates the 95% significance level by t-test and the Benjamini-Hochberg procedure. (e) Histogram of 1980:2015 September-October mean heat flux over the WSP region (red box in Fig. 1) for the years 1980 through 2015. A best fit normal distribution is shown in solid red line while a pink star denotes the 1974 value (f) as (e) but for surface temperature. (g) as (e) but for MSLP. (h) as (e) but for precipitation.**