# Peer review of "A comparison of the atmospheric response to the Weddell Sea Polynya in AGCMs of varying resolutions."

_EGUsphere, 2023_

## Referee Comment (RC1)

A Review of the Manuscript

**"A comparison of the atmospheric response to the Weddell Sea Polynya in AGCMs of varying resolutions."**

Holly. C. Ayres[1], David. Ferreira[1], Wonsun. Park[2,3,4],
Joakim. Kjellsson[2,5], Malin. Ödalen[2]

[1] Department of Meteorology, University of Reading, Reading, UK, [2] Division of Ocean Circulation and Climate Dynamics, GEOMAR Helmholtz Centre for Ocean Research Kiel, Germany, [3] IBS Center for Climate Physics, Institute for Basic Science (IBS), Busan, Republic of Korea, [4] Department of Climate System, Pusan National University, Busan, Republic of Korea, [5] Kiel University, Kiel, Germany

This study evaluates the atmospheric response to a Weddell Sea Polynya (WSP) in the austral winter of 1974 using a somewhat idealized experimental setup of AGCMs at low- and high-resolutions. The experimental setup of the present study tries to isolate the response of a WSP by looking at the differences between atmospheric responses in experiments with and without WSP by modifying the boundary conditions in ERA5 sea ice and SST fields. Their findings show that the atmospheric responses in all the simulations are restricted to the immediate vicinity of the WSP in the horizontal and boundary layer in the vertical, primarily driven by the ocean to atmospheric turbulent heat flux. The impact of changing resolution is seen in the magnitude of the response at low levels, with high-resolution simulations having a more significant response. These results are expected, given that Weijer et al. 2017 reached similar conclusions using a fully-coupled GCM with an atmospheric model component resolution of 25km.

Major Comments:

The study has mainly suffered from the experimental setup since the varying resolutions apply only to the atmosphere. However, the ocean and sea ice forcing data remain the same in resolution (ERA5 data standard 31km horizontal resolution). It should have been evident to the authors that given the numerous high-resolution studies on the mechanisms of WSPs, to look at just the atmospheric response of the WSPs, one would need to use atmospheric models much higher than 25 km horizontal resolution despite the size of the WSP (the great WSPs of 1974-1976 remained ice-free throughout the austral winter with an ice-free region of ~ 250,000 sq. km). It would have been valuable for the authors also to consider high-resolution fully-coupled modeling studies that have simulated and investigated the cause/effects of realistic open ocean polynyas in the Southern Ocean (Dufour et al. 2017; Kaufman et al. 2020; Gutjahr et al. 2018; Stössel et al. 2015; Chang et al. 2020; Kurtakoti et al. 2021; Weijer et al.). It might help the authors to see how the representation of the WSP changes with resolution in coupled climate simulations and how the atmosphere responds to the WSPs. In those above fully-coupled studies investigating open ocean polynyas, Weijer et al. 2017 exclusively looked at the atmospheric response to WSPs, and their atmospheric model component was configured at 25km horizontal resolution. To improve the manuscript, the authors may want to investigate how the clouds characteristics, radiative fluxes change and/or modify the cloud radiative effects over the WSP in these simulations.

Minor Comments

Lines 18-20:  Please explain this further.

Line 34: "perhaps having only occurred once per century".

The exact frequency of WSP in the past is not known. Studies have shown a strong link between the southern hemisphere westerlies, SAM index and WSP formation (Cheon et al. 2017; Gordon et al. 2007; Gordon 2014). Gordon et al., 2007 mentions "Gordon (1982) reports that two hydrographic stations obtained by the Argentine ship San Martin in 1961 reveal the absence of the warm deep water, similar to conditions encountered in the 1977 Islas Orcadas stations. The SAM index indicates a prolonged negative SAM in the decade prior to the possible polynya in the winter of 1960. Furthermore, except for the 5-yr period centered on 1910, a negative or neutral SAM index persisted from the 1890s into the first three decades of the twentieth century. Might the Weddell Polynya have been common then?"

References

Chang, P., and Coauthors, 2020: An Unprecedented Set of High-Resolution Earth System Simulations for Understanding Multiscale Interactions in Climate Variability and Change. J Adv Model Earth Syst, 12, e2020MS002298, https://doi.org/10.1029/2020MS002298.

Cheon, W. G., and Coauthors, 2017: The Role of Oscillating Southern Hemisphere Westerly Winds: Southern Ocean Coastal and Open-Ocean Polynyas. J Clim, JCLI-D-17-0237.1, https://doi.org/10.1175/JCLI-D-17-0237.1.

Dufour, C. O., A. K. Morrison, S. M. Griffies, I. Frenger, H. Zanowski, and M. Winton, 2017: Preconditioning of the Weddell Sea polynya by the ocean mesoscale and dense water overflows. J Clim, 7719–7737, https://doi.org/10.1175/JCLI-D-16-0586.1.

Gordon, A. L., 2014: Southern Ocean polynya. Nat Clim Chang, 4, 249–250.

Gordon, A. L., M. Visbeck, and J. C. Comiso, 2007: A possible link between the Weddell Polynya and the southern annular mode. J Clim, 20, 2558–2571, https://doi.org/10.1175/JCLI4046.1.

Gutjahr, O., D. Putrasahan, K. Lohmann, J. H. Jungclaus, J.-S. von Storch, N. Brüggemann, H. Haak, and A. Stössel, 2018: Max Planck Institute Earth System Model (MPI-ESM1.2) for High-Resolution Model Intercomparison Project (HighResMIP). Geoscientific Model Development Discussions, 1–46, https://doi.org/10.5194/gmd-2018-286.

Kaufman, Z. S., N. Feldl, W. Weijer, and M. Veneziani, 2020: Causal interactions between southern ocean polynyas and high-latitude atmosphere-ocean variability. J Clim, 33, 4891–4905, https://doi.org/10.1175/JCLI-D-19-0525.1.

Kurtakoti, P., M. Veneziani, A. Stössel, W. Weijer, and M. Maltrud, 2021: On the generation of Weddell Sea polynyas in a high-resolution earth system model. J Clim, 34, 2491–2510, https://doi.org/10.1175/JCLI-D-20-0229.1.

Stössel, A., D. Notz, F. A. Haumann, H. Haak, J. Jungclaus, and U. Mikolajewicz, 2015: Controlling High-Latitude Southern Ocean Convection in Climate Models. Ocean Model (Oxf), 86, 58–75, https://doi.org/10.1016/j.ocemod.2014.11.008.

Weijer, W., M. Veneziani, M. W. Hecht, N. Jeffery, and A. Jonko, Atmospheric Response to the Weddell Sea Polynya.

---

## Referee Comment (RC2)

**Review of 'A comparison of the atmospheric response to the Weddell Sea Polynya in AGCMs of varying resolutions'**

**The Cryosphere**

Holly. C. Ayres[1], David. Ferreira[1], Wonsun. Park[2,3,4], Joakim. Kjellsson[2,5], Malin. Ödalen[2]
[1] Department of Meteorology, University of Reading, Reading, UK
[2] Division of Ocean Circulation and Climate Dynamics, GEOMAR Helmholtz Centre 5 for Ocean Research Kiel, Germany
[3] IBS Center for Climate Physics, Institute for Basic Science (IBS), Busan, Republic of Korea
[4] Department of Climate System, Pusan National University, Busan, Republic of Korea
[5] Kiel University, Kiel, Germany
*Correspondence to*: Holly Ayres h.c.ayres@reading.ac.uk

The objective of this paper is to assess the impact of the Weddell Sea Polynya (WSP) on the atmosphere via the ocean-air transfer of heat during the opening in the 70s. The method used by the authors is based on modelling using 3 atmospheric models with prescribed sea ice and SSTs. The scientific question is interesting however the methodology is not valid and I invite the authors to rethink how they can improve their strategy to investigate this question.

Major comments:

- The authors could have used satellite data for sea ice instead of ERA5 in order to get an accurate coverage of sea ice during the polynya event.

- The authors interpret the anomalies in heat, temperature and precipitation as being due to the polynya. However, it has been shown that during the polynya events (2017 and 70s) there is an excess of heat and precipitation coming from the atmospheric rivers **toward** to ocean. How the authors can be sure that the values they obtained for the different parameters are solely due to ocean-to-air transfer of heat and not to the existing atmospheric conditions (atmospheric rivers and cyclones) which lasted for several days? This is critical and needs to be addressed by the authors perhaps by conducting sensitivity studies using the models and comparing one set of simulations **with** sea ice opening and one **without** sea ice opening but both with the same atmospheric conditions i.e. those occurring during the polynya events.

- During the 2017 event there are in-situ measurement from the SOCCOM network that can be used at least to check how the models are performing.

---

## Author Comment (AC1)

We thank the reviewers for their helpful feedback on our manuscript. We address both of the reviewers' concerns and suggestions below. We have included clarifications and changes to the manuscript.

**Review 1**

Major comments:

'These results are expected, given that Weijer et al. 2017 reached similar conclusions using a fully-coupled GCM with an atmospheric model component resolution of 25km.' and 'In those above fully-coupled studies investigating open ocean polynyas, Weijer et al. 2017 exclusively looked at the atmospheric response to WSPs, and their atmospheric model component was configured at 25km horizontal resolution.'

We agree with the reviewer that (1) our results on the response of the atmosphere to the polynya are in line with expectations although this is due in part to the reviewer's interpretation that (2) Weijer et al.'s work is an investigation of the direct atmospheric response to the polynya, which we do not agree with. These 2 points are developed below.

- (1) Indeed, our findings are somewhat expected. However, we argue that in itself this does not make them unworthy of publication. These expectations will remain speculations until the proper experiments are carried out to confirm expectations. This is the main goal of our study. Note that the direct response of the atmosphere to the polynya has not been explored since the studies of Dare and Atkinson (1999, 2000) and Timmermann et al. (1999). However, Dare and Atkinson did not use a full 3dimensional Global Circulation Model, but a 2-dimensional (x,z) model spanning the boundary layer in height (1000 m) and 100 km across the polynya. Timmermann et al. (1999) assumed a warming in response to the polynya and inferred the atmospheric circulation response from thermal wind balance assuming a level of no-motion at 6000 m height. Our study using 3 GCMs at 2 resolutions each is a much extended investigation compared with previous studies, providing a 3-D, global, fully dynamical estimation of the atmospheric response to the polynya, as well as a test of the robustness of the results to model and resolution dependencies. To our knowledge, this is the most extensive AGCM study conducted for the response to the Weddell Sea Polynya (WSP).
- (2) The synchronous composite analysis carried out by Weijer et al. (2017) in a coupled model does not allow to separate cause and effect. Without further information, the atmospheric signal they extracted cannot be interpreted as the direct response of the atmosphere to the polynya, but as a mix of this direct response and feedbacks. In fact, Diao et al. (2022), who used the same model as Weijer et al., argued that the polynya is the result of a coupled ocean-atmosphere mechanism. For similar reasons, the atmospheric signal extracted by Moore et al. (2002) from the NCEP-NCAR reanalysis which integrates observations of the real atmosphere and hence potential secondary feedbacks, cannot be either interpreted as the direct response of the atmosphere to the polynya. An additional issue (which does not directly pertain to the reviewer's comment but feeds into the novelty of our work) is that both Weijer et al. and Moore et al. estimate the atmospheric signal from a very small sample of polynya realisations (3)

and 1, respectively), so there are large uncertainties due to the imprint of internal variability.

To separate the direct atmospheric response from the potential feedbacks in coupled models, one needs to use additional uncoupled (AGCM) simulations where feedbacks are disallowed. A recent study by the lead author highlights the need for AGCM experiments, with coupled models, in order to isolate the direct response from coupled feedbacks (Ayres et al., 2022). The use of AGCM experiments with prescribed surface boundary conditions is a standard approach which has been extensively used in the mid-latitudes and polar regions to investigate coupled interactions. As argued in point (1), our study addresses this point with state-of-the art models providing a very significant improvement compared to the previous literature.

A posteriori, in light of our results, it appears that the atmospheric signal extracted by Weijer et al. is to the first order of the direct response to the polynya, and that if there are coupled feedbacks in their model, they are small. Our results suggest that feedbacks invoked by Diao et al., are weak.

The reviewer's comment made us realise that we have not made a good job of explaining the scope of the paper. We specifically intend to *not* use coupled models.

We have made our goals clearer in the manuscript, stating our differences to the Wiejer et al paper, and how this work complements theirs.

'The study has mainly suffered from the experimental setup since the varying resolutions apply only to the atmosphere.'

With the clarifications above about the scope of our study, we now hope it is clear that the varying resolution being applied to only the atmosphere is not a limitation but a goal of our study.

Again, we emphasise that the purpose of our study is to infer the direct atmospheric response to the WSP through atmosphere-only models, avoiding the ambiguities around causality and feedbacks that arise in coupled models. Our study complements and helps the interpretation of the coupled model studies that assess the full atmospheric signal associated with the polynya.

By applying a boundary forcing of sea surface temperature (SST) and sea ice to the model, we infer the direct response with no secondary feedback with the ocean, a method used to infer the atmospheric response to SST and sea ice change for many decades, dating back to 1990 with (e.g., Royer et al., 1990), to the present day (e.g., Zheng et al., 2023).

We use this method because we do not have many Weddell Sea polynyas in our satellite record history, and many polynyas that occur in coupled models are not realistic (i.e., too big or occur too frequently, with poor ocean representation). We apologise if we have not made this clear in the text, and have now emphasised this further in the manuscript.

'However, the ocean and sea ice forcing data remain the same in resolution (ERA5 data standard 31km horizontal resolution). It should have been evident to the authors that given the numerous high-resolution studies on the mechanisms of WSPs, to look at just the atmospheric response of the WSPs, one would need to use atmospheric models much higher than 25 km horizontal resolution despite the size of the WSP (the great WSPs of 1974-1976 remained ice-free throughout the austral winter with an ice-free region of ~ 250,000 sq. km).'

As discussed in the first paragraph of the methods and data section, the resolution of the boundary conditions makes little difference to the response. There is limited accurate satellite data for the 1974 WSP, hence our use of ERA5, which is one of the most accurate reanalysis products available. When compared to other reanalysis, ERA5 demonstrates the best performance in representing many processes over Antarctica (e.g., Gossart et al. 2019). We do mention limitations of ERA5 in the text, which we apologise is not clear, and have made changes accordingly.

During our initial sensitivity experiments we tested our model runs with the 2017 Maud Rise polynya (MRP) satellite data (Merchant et al. 2019), and compared it to the 2017 ERA5 forcing. The satellite and ERA5 reanalysis showed little difference for the 2017 Polynya with up to 0.1 K SST difference under sea ice, and ~0.5 K difference in the polynya region (Fig. R1). SIC within the polynya shows no difference between the two datasets. The higher ERA5 SST produces a slightly greater heat flux but we found no meaningful difference to our results beyond this.

Satellite data was not available for 1974, therefore a comparison was not possible for this event. We chose the 1974 WSP to maximise the signal-to-noise ratio due to the larger polynya that year, and the literature discussion (e.g., Cheon and Gordon 2019), that due to the smaller size and position of the 2017 MRP (80,000 km2), it is not considered as a WSP.

In ERA5, the 1974 data is of slightly worse quality than data after 1979 as a consequence of the limited satellite observations at the time. Crucially, for this study, high accuracy of SIC and SST is not critical- that is we do not expect that change of these boundary conditions, that would impact a few in grid points bordering the polynya, will significantly affect the atmospheric response.

Additionally, for our model resolutions, an objective of this study is to explore the impact of the polynya at varying resolution AGCMs, which to our knowledge has not been done before. We could not find in the suggested coupled modelling papers by the reviewers a clear statement that the atmospheric response to the surface boundary conditions should be carried out exclusively "at much higher resolution than 25 km". While we agree that higher resolution would be better (although this is all subject to debate), our experiments are at the forefront of what is routinely feasible. Our experiments with HadGEM3 and OpenIFS at high resolution use atmospheric resolution of about 25 km.

On a more pragmatic level, we do not have access to a global well-tuned atmospheric model with much higher resolution than 25 km that we could afford to run on large ensemble.

Figure R1: An experiment conducted at the start of the study, to determine the differences between satellite and ERA5 sea ice and SSTs for the 2017 Maude Rise polynya.

'It would have been valuable for the authors also to consider high-resolution fully coupled modelling studies that have simulated and investigated the cause/effects of realistic open ocean polynyas in the Southern Ocean (Dufour et al. 2017; Kaufman et al. 2020; Gutjahr et al. 2018; Stössel et al. 2015; Chang et al. 2020; Kurtakoti et al. 2021; Weijer et al.). It might help the authors to see how the representation of the WSP changes with resolution in coupled climate simulations and how the atmosphere responds to the WSPs.'

Most of the studies listed by the reviewer focus on the preconditioning and formation mechanism of the polynya, which is not the topic of our research. We are concerned by the impact of the polynya on the atmosphere once it is formed. Some of the suggested papers discuss the atmospheric signal associated with the polynya. However, as discussed in the point (1) above, there are ambiguities in interpreting the simultaneous atmospheric signal in a coupled model as the atmospheric response to the polynya. Note that this is not impossible but one needs to use sophisticated analysis techniques such as the lagged maximum covariance analysis to disentangle cause and effects (see for example, Czja and Frankignoul 2002, for an attempt to extract the atmospheric response to SST in the mid-latitudes).

Additionally, there are no models that we know of, in which WSPs are represented systematically across varying models and resolutions, where most of the suggested studies come from the same model and research group. Therefore, a direct comparison to the suggested literature is not feasible and is irrelevant to our study. Additionally, where relevant, we have already discussed these models in both the introduction, and the discussion where we discuss the potential feedbacks that these coupled models induce. We may not have emphasised enough that we aim to compliment the coupled models with uncoupled models. We have included some additional information on these studies in our manuscript.

To improve the manuscript, the authors may want to investigate how the clouds characteristics, radiative fluxes change and/or modify the cloud radiative effects over the WSP in these simulations.

We thank the reviewer for the suggestion. Unfortunately, we did not save any outputs directly relating to cloud characteristics. However, we do have net top of atmosphere (TOA) shortwave and longwave fluxes. We show below the response of these two fields to the WSP in the OIFS-HR model. The other models show similar patterns and magnitudes.

The net TOA shortwave response to the WSP is shown in Fig. R2 (left). There is an increase in net shortwave absorption local to the polyna, as expected when reducing the sea ice in the region. The net TOA longwave radiation response (Fig. R2, right) shows a small decrease (~ -8 Wm-2), i.e. a small increase in outgoing long wave radiation. This could reflect a small increase in the cloud-top height. However the longwave effect is dwarfed by the shortwave effect, i.e. the sea ice albedo effect.

Note that we show in Fig. R2 the September-October average of the fluxes anomalies. The shortwave effect is in fact highly time dependent, varying from near zero in August (no incoming sunlight) to a maximum in November (start of the summer).

We ask the reviewer if they would suggest us including this analysis in the final manuscript?

Figure R2: (left) September - October mean top of atmosphere shortwave flux response to the WSP for OIFS-HR (polynya - non-polynya). (right) as left for longwave. Positive in the down direction.

**Minor Comments:**

Lines18-20:Please explain this further.

We thank the reviewer for highlighting that this is unclear, we have adjusted the manuscript accordingly.

Line 34: "perhaps having only occurred once per century".

The exact frequency of WSP in the past is not known. Studies have shown a strong link between the southern hemisphere westerlies, SAM index and WSP formation (Cheon et al. 2017; Gordon et al. 2007; Gordon 2014). Gordon et al., 2007 mentions "Gordon (1982) reports that two hydrographic stations obtained by the Argentine ship San Martin in 1961 reveal the absence of the warm deep water, similar to conditions encountered in the 1977 Islas Orcadas stations. The SAM index indicates a prolonged negative SAM in the decade prior to the possible polynya in the winter of 1960. Furthermore, except for the 5-yr period centred on

1910, a negative or neutral SAM index persisted from the 1890s into the first three decades of the twentieth century. Might the Weddell Polynya have been common then?

We thank the reviewer for these interesting suggestions but we think that such speculations on the frequency of the polynya would take the reader on a tangent and weaken our core message. The aim of our study is not the formation mechanism and frequency of polynya, but to extract the atmospheric response to the polynya to set up a solid basis on which to interpret coupled simulations and observations. Indeed we hope that our work will help with pushing further the suggestions made above.

Returning to the frequency of the polynya, we prefer to stir away from this debate and we will replace this statement by a more generic statement relying on published literature.

**Review 2:**

The methodology is not valid and I invite the authors to rethink how they can improve their strategy to investigate this question.

We are unsure as to why the reviewer thinks that our method is not valid, or which aspect specifically. The statement that the method is invalid is not followed by any explanation. The method used in this paper is rather standard and has been used to explore the direct response to sea ice change with the use of sea ice concentration and sea surface temperature in the polar region from back to 1990 (e.g., Royer et al., 1990) to the present day (e.g., Zheng et al., 2023).

Possibly, the reviewer misunderstood the aims of the study and hence the approach. We hope that the clarifications below help dissipate this potential misunderstanding.

**- The authors could have used satellite data for sea ice instead of ERA5 in order to get an accurate coverage of sea ice during the polynya event.**

The reviewer suggests using satellite data for our experiment, however, there is no gridded monthly satellite data for the 1974 polynya, which is required for our boundary conditions. If they mean for us to use satellite data for the 2017 MRP, then that is not the purpose of this study as the 2017 polynya was not classified as a WSP and is smaller than the 1974 polynya (hence issues with the signal to noise ratio).

Therefore, for the aim of this study, we could not have used satellite data, and ERA5 is the most accurate coverage we have. Note that the ERA5 sea ice concentration field is a merged product of available data (satellite, in-situ); whatever satellite data are available for 1974 would be included in the ERA5 sea ice.

- The authors interpret the anomalies in heat, temperature and precipitation as being due to the polynya. However, it has been shown that during the polynya events (2017 and 70s) there is an excess of heat and precipitation coming from the atmospheric rivers **toward** to ocean. How the authors can be sure that the values they obtained for the different parameters are solely due to ocean-to-air transfer of heat and not to the existing atmospheric conditions (atmospheric rivers and cyclones) which lasted for several days? This is critical and needs to be addressed by the authors perhaps by conducting sensitivity studies using the models and comparing one set of simulations **with** sea ice opening and one **without** sea ice opening but

both with the same atmospheric conditions i.e. those occurring during the polynya events.

Our study uses an ensemble method, where the state of the atmosphere is different each year, and is averaged with statistical significance applied. We run simulations with and without the polynya in an atmospheric GCM, by design, our results (polynya minus non-polynya simulations) only show the direct atmospheric response to the polynya. This is an extremely common method for modelling studies.

We apologise that we must not have emphasised the method of ensemble averaging and the use of monthly means, in addition to significance testing to reduce the impacts of short-lived weather events. Additionally, the use of AGCMs would mean that there are no impacts of these short-lived (or any) events on the ocean or polynya, that is in fact the reasoning for our method, to eliminate secondary ocean feedbacks that one would get in a coupled model. We will make these points clearer in the manuscript.

The reviewer's suggestion of an additional computations with prescribed sea ice and atmospheric state effectively reduces to a bulk formula calculation of the fluxes (everything else would be prescribed). Effectively, the results of such a computation would be close to that of the ERA5 fluxes where the sea ice and SST are prescribed, and the atmospheric state is constrained (but not prescribed) by observations. In fact, to carry out such a computation in practice, one would need a gridded atmospheric product for surface temperature, humidity, winds etc, and thus would turn to a reanalysis product such as ERA5. In this case, the suggested computation would return exactly the air-sea/air-ice fluxes of ERA5 presented in our Fig. 8.

More importantly, we believe that such a computation would not answer the reviewer's question about causality. In addition, we are not concerned here by the effect of the atmosphere on the polynya but on the effect of the polynya and the atmosphere.

- During the 2017 event there are in-situ measurement from the SOCCOM network that can be used at least to check how the models are performing.

This suggestion is irrelevant to our study for multiple reasons. The first being that our study is based on the 1974 WSP (250,000 km2), not the 2017 MRP (80,000 km2), which are two different phenomena with very different in size, and thus the heat flux from the ocean to the atmosphere would be very different (Moore et al., 2002).

Second, as far as we are aware the SOCCOM data for this 2017 polynya are only ocean data. We do not have an ocean component in our model. We prescribed the ocean surface state to infer the atmospheric response. Possibly, the reviewer suggests that we use the SST sampled by SOCCOM to constraint the prescribed SST in our experiments. However, the SOCCOM sampling is extremely sparse in space and time, while we need a gridded SST field. This would be a limited data set to draw on for the atmospheric response to the polynya.

Finally, we do in fact validate our models with ERA5 data in section 3.4. Here we use ERA5 data for two main purposes, one, give a direct comparison based on our model input data, and

two, as mentioned in an earlier comment, despite the limitations of using reanalysis, ERA5 is one of the best products we have for the 1974 polynya event.

**References**

Ayres, H. C., Screen, J. A., Blockley, E. W., & Bracegirdle, T. J. (2022). The Coupled Atmosphere–Ocean Response to Antarctic Sea Ice Loss. Journal of Climate, 35(14), 4665–4685. https://doi.org/10.1175/JCLI-D-21-0918.1

Cheon, W. G., & Gordon, A. L. (2019). Open-ocean polynyas and deep convection in the Southern Ocean. Scientific Reports, 9(1), 1–9. https://doi.org/10.1038/s41598-019-43466-2

Czaja, A., & Frankignoul, C. (2002). Observed Impact of Atlantic SST Anomalies on the North Atlantic Oscillation. https://doi.org/https://doi.org/10.1175/1520-0442(2002)015%3C0606:OIOASA%3E2.0.CO;2

Dare, R. A., & Atkinson, B. W. (1999). Numerical modeling of atmospheric response to polynyas in the Southern Ocean sea ice zone. Journal of Geophysical Research Atmospheres, 104(D14), 16691–16708. https://doi.org/10.1029/1999JD900137

Diao, X., Stössel, A., Chang, P., Danabasoglu, G., Yeager, S. G., Gopal, A., Wang, H., & Zhang, S. (2022). On the Intermittent Occurrence of Open-Ocean Polynyas in a Multi-Century High-Resolution Preindustrial Earth System Model Simulation. Journal of Geophysical Research: Oceans, 127(4). https://doi.org/10.1029/2021JC017672

Gossart, A., Helsen, S., Lenaerts, J. T. M., Vanden Broucke, S., van Lipzig, N. P. M., & Souverijns, N. (2019). An evaluation of surface climatology in state-of-the-art reanalyses over the Antarctic Ice Sheet. Journal of Climate, 32(20), 6899–6915. https://doi.org/10.1175/JCLI-D-19-0030.1

Merchant, C. J., Embury, O., Bulgin, C. E., Block, T., Corlett, G. K., Fiedler, E., Good, S. A., Mittaz, J., Rayner, N. A., Berry, D., Eastwood, S., Taylor, M., Tsushima, Y., Waterfall, A., Wilson, R., & Donlon, C. (2019). Satellite-based time-series of seasurface temperature since 1981 for climate applications. Scientific Data, 6(1), 1–18. https://doi.org/10.1038/s41597-019-0236-x

Moore, G. W. K., Alverson, K., & Renfrew, I. A. (2002). A reconstruction of the air-sea interaction associated with the Weddell polynya. Journal of Physical Oceanography, 32(6), 1685–1698. https://doi.org/10.1175/1520-0485(2002)032<1685:AROTAS>2.0.CO;2

Royer, J. F., Planton, S., & D~qu~, M. (1990). limud¢ Dynamics A sensitivity experiment for the removal of Arctic sea ice with the French spectral general circulation model\*. In Climate Dynamics (Vol. 5). https://doi.org/https://doi.org/10.1007/BF00195850

Weijer, W., Veneziani, M., Stössel, A., Hecht, M. W., Jeffery, N., Jonko, A., Hodos, T., & Wang, H. (2017). Local atmospheric response to an open-ocean polynya in a highresolution climate model. Journal of Climate, 30(5), 1629–1641. https://doi.org/10.1175/JCLI-D-16-0120.1

Timmermann, R., Lemke, P., & Kottmeier, C. (1999). Formation and Maintenance of a Polynya in the Weddell Sea.

Zheng, C., Y. Wu, M. Ting, J. A. Screen, and P. Zhang, 2023: Diverse Eurasian Temperature Responses to Arctic Sea Ice Loss in Models due to Varying Balance between Dynamic Cooling and Thermodynamic Warming. J. Climate, 36, 8347–8364, https://doi.org/10.1175/JCLI-D-22-0937.1.

---

## Referee Report (RR1)

**Review of *"A comparison of the atmospheric response to the Weddell Sea Polynya in AGCMs of varying resolutions."**

*Holly. C. Ayres[1], David. Ferreira[1], Wonsun. Park[2,3,4], Joakim. Kjellsson[2,5], Malin. Ödalen[2]*
*[1] Department of Meteorology, University of Reading, Reading, UK*
*[2] Division of Ocean Circulation and Climate Dynamics, GEOMAR Helmholtz Centre for Ocean Research Kiel, Germany*
*[3] IBS Center for Climate Physics, Institute for Basic Science (IBS), Busan, Republic of Korea*
*[4] Department of Climate System, Pusan National University, Busan, Republic of Korea*
*[5] Kiel University, Kiel, Germany*
*Correspondence to: Holly Ayres h.c.ayres@reading.ac.uk*

The authors have conducted a study which identifies the direct response of the atmosphere to the 1974 Weddell Sea Polynya (WSP) in Atmospheric General Circulation Models (AGCMs), and compares this to an analysis of the response in the ECMWF Reanalysis version 5 (ERA5). By prescribing ERA5 sea surface temperature (SST) and sea ice concentration (SIC) as surface boundary conditions for the AGCMs, the authors explore the range of direct responses across three different dynamical models, and the response across two resolutions for each model.

Previous coupled simulations have shown an importance of model resolution in their representation of Southern Ocean polynyas (e.g. Lockwood et al. 2021), however there is limited research isolating the model response (as opposed to formation combined with response), and understanding how this varies with model resolution. Therefore understanding the effect of model resolution on the atmospheric response to polynyas is a valuable field of study. Comparing the direct atmospheric response to a reanalysis response can also provide some insight on the pathways for a climate response to the WSP. Finally, the study of the reanalysis response itself, with the state-of-the-art reanalysis model in ERA5 is also valuable in understanding the real-world response to WSPs.

Summarising the above paragraph, in my opinion the three main valuable contributions of this research paper are as follows:
- Exploring how model resolution impacts the atmospheric response to the WSP.
- Understanding the atmospheric-only response, and how this compares to the reanalysis response to provide hints on the relative importance of the direct atmospheric response with respect to responses which include coupled feedbacks.
- Analysing the ERA5 reanalysis response to the WSP.

I also would like to recognise that although the results of this study are primarily not unexpected or conceptually new, I still consider it a valuable contribution to repeat methodologies with state-of-the art models. Additionally, this research extends previous studies with a multi-model, multi-resolution framework to provide a comprehensive study of the direct atmospheric response to the WSP.

Suggested revisions:

In my opinion, the authors need to take care that the wording of conclusory remarks remains within the scope of this direct atmosphere-only response study.

For example, the authors' statement *"...our results suggest that a 1974-size polynya (the largest observed) cannot generate a response that would project on the scale of the Weddell gyre."* is too much of an extrapolation, the study is isolated to atmosphere-only dynamics, and so can not alone rule out wider responses which could arise from coupled feedbacks.

Similarly, I think the following statement should be reworded, its wording implies that the results of this study prohibit a large-scale interaction of the WSP with the climate, whereas in reality, the results only suggest that because the reanalysis results and atmosphere-only results are similar, that

the influence of the coupled feedbacks on the atmosphere are small. The WSP may influence wider ocean circulation, which could in turn have climate impacts elsewhere, and this possibility cannot be ruled out by this study alone: *"...the WSP may not interact with the climate on a large scale and may be too far south and within sea ice edge for coupled feedbacks to have a substantial impact on results".*

Again in the statement *"In addition, our results show that the response has little memory (i.e., local in time too) and vanishes rapidly with the polynya",* the authors need to clarify that this means that there is little memory in the atmospheric-only system. Memory in a climate system is often established via ocean-atmosphere-sea ice feedbacks, which by design are not present in this study.

The statement on line 94, *"The interpretation of coupled models is made difficult by the potential impact of ocean-atmosphere-sea ice feedbacks",* needs to be justified. A reader might argue that a coupled model is easier to interpret, because it includes feedbacks which an atmosphere-only model does not, and the processes which drive these feedbacks exist in the real-world. Comments such as this, and as those quoted above, present the idea that these direct atmosphere-only simulations are a better alternative to coupled simulations, when in fact these different methodologies answer different scientific questions.

The authors have discussed the fact that they are using ERA5 before the assimilation of satellite data, and commented on potential data quality issues. They have also cautioned the reader that the ERA5 data is the result of one realisation. I would therefore strongly recommend that the authors utilise all ten members of the ERA5 reanalysis to improve their analysis of the response to the polynya in ERA5, and to provide uncertainty estimates on this response. ERA5 is likely weakly observationally constrained in the Southern Ocean, particularly in the pre-satellite era, which means that the underlying model may play a significant role in determining the reanalysis conditions in the area of interest.

With regards to the statistical methodology, I would suggest that the authors consider employing multiple-hypothesis testing when applying the students t-test to potentially spatially correlated data (e.g. surface temperature and precipitation response). As described by Wilks (2006), this is a mistake that is made by standard in the field. In my opinion, at least acknowledging this potential statistical caveat is an important step in improving the statistical robustness in our community.

The paper is generally well written and presented, but could also benefit from a proofread. I have included below a list of some of the minor errors which I noticed, but I did not have the time to go through the entire article from a spelling and grammar point of view:

Spelling mistake in line 16 "isolate" should be "isolates".
Spelling mistake in line 68 "impact" should be "impacts".
Spelling mistake in line 112 "your" should be "our".
Line 121 "...thus, has limitations" the reason for limitations needs to be clarified a little.
Line 180 "limiting".
Authors should decide to use "sea ice" or "sea-ice" but not both.

Finally, I would like to thank the authors for their contribution, which is a very interesting research topic.

Kind regards,
Tarkan A Bilge

Lockwood, J.W., Dufour, C.O., Griffies, S.M., and Winton, M.: On the role of the Antarctic Slope Front on the occurrence of the Weddell Sea polynya under climate change, J. Climate, 34, 1–56, 2021.

Wilks, D. S., 2006: On "Field Significance" and the False Discovery Rate. _J. Appl. Meteor. Climatol._, 45, 1181–1189,
[https://doi.org/10.1175/JAM2404.1](https://doi.org/10.1175/JAM2404.1).

---

## Referee Report (RR2)

A Review of the resubmitted Manuscript
**"A comparison of the atmospheric response to the Weddell Sea Polynya in AGCMs of varying resolutions."**

Holly. C. Ayres[1], David. Ferreira[1], Wonsun. Park[2,3,4],
Joakim. Kjellsson[2,5], Malin. Ödalen[2]

[1] Department of Meteorology, University of Reading, Reading, UK, [2] Division of Ocean Circulation and Climate Dynamics, GEOMAR Helmholtz Centre for Ocean Research Kiel, Germany, [3] IBS Center for Climate Physics, Institute for Basic Science (IBS), Busan, Republic of Korea, [4] Department of Climate System, Pusan National University, Busan, Republic of Korea, [5] Kiel University, Kiel, Germany

I appreciate the authors' effort to address some of the issues mentioned in the earlier manuscript review. Most of the authors' rewriting efforts greatly help better understand the results and discussion section of the paper. However, that alone does not recommend publication, considering that, as the authors themselves state, "the results are somewhat expected." The authors have presented no new analysis other than what was in the original submission. The manuscript does not have enough analysis to present a well-rounded story that would do justice to the goal of the manuscript. The authors do not save any of the outputs related to cloud characteristics, which is strange considering they wanted to study the atmospheric response. Previous studies that look at the atmospheric response to a Weddell Sea Polynya have found that most of the atmospheric response is limited to the vicinity of the open ocean polynya in terms of cloud and precipitation changes. Thus, the authors would have wanted to save the atmospheric variables that quantify the cloud characteristics and analyze the difference in cloud and precipitation characteristics with changing resolution. Furthermore, it is justifiable that at least one of the simulations should have a higher spatial resolution of ~0.1°.

---

## Referee Report (RR3)

**Second review of *"A comparison of the atmospheric response to the Weddell Sea Polynya in AGCMs of varying resolutions."**

*Holly. C. Ayres[1], David. Ferreira[1], Wonsun. Park[2,3,4], Joakim. Kjellsson[2,5], Malin. Ödalen[2]*
*[1] Department of Meteorology, University of Reading, Reading, UK*
*[2] Division of Ocean Circulation and Climate Dynamics, GEOMAR Helmholtz Centre for Ocean Research Kiel, Germany*
*[3] IBS Center for Climate Physics, Institute for Basic Science (IBS), Busan, Republic of Korea*
*[4] Department of Climate System, Pusan National University, Busan, Republic of Korea*
*[5] Kiel University, Kiel, Germany*
*Correspondence to: Holly Ayres h.c.ayres@reading.ac.uk*

I would like to thank the authors for their thorough response to my first review, and I am glad to see that they have incorporated all my previous suggestions. The authors have reworded and clarified statements regarding the impact of the study, applied a method to control the False Detection Rate, and utilised all ten members of ERA5 reanalysis. I would also like to thank the authors' for their further explanation of the significance of their study in the context of other recent research.

I now just have a couple of minor suggestions for the authors in light of their revisions:

- It would be good to include the reference for the Benjamini-Hochberg method in your bibliography.
- The author has written "...by t-test and False Discovery Rate" in a number of places, it would be more accurate to replace "False Discovery Rate" in this context by "The Benjamini-Hochberg procedure". Technically, the False Discovery Rate is the rate at which statistical tests falsely appear significant, and the Benjamini-Hochberg procedure is a method to control the FDR (i.e. it is a "False Discovery Rate controlling procedure").
- Is there a version number that accompanies HadGEM3? I thought that HadGEM3 was the name of the whole model family - perhaps you could check this, I could be wrong here. For example, perhaps it is called HadGEM3-AO. The reference given (Williams et al. 2017) doesn't seem to be in your bibliography.
- In my opinion, the point in line 94 "The interpretation of coupled models and of the direct response to the WSP is made difficult..." should be reworded to "Interpreting the direct response of the WSP in coupled models is made difficult..." or similar - at the moment it does not quite work grammatically.
- Could you double-check the bibliography, I have noticed a couple of missing entries (above) but also some of the entries are incomplete, e.g. "Timmermann, R., P. Lemke, and C. Kottmeier, 1999: Formation and Maintenance of a Polynya in the Weddell Sea." has no journal name.
- Typo in Figure 6 caption; "solid".

Kind regards,
Tarkan A Bilge

---

## Author Response (AR2)

We thank the reviewers for their helpful feedback on our manuscript. We address both of the reviewers' concerns and suggestions below. We have included clarifications and changes to the manuscript.

Review 1:

The authors have conducted a study which identifies the direct response of the atmosphere to the 1974 Weddell Sea Polynya (WSP) in Atmospheric General Circulation Models (AGCMs), and compares this to an analysis of the response in the ECMWF Reanalysis version 5 (ERA5). By prescribing ERA5 sea surface temperature (SST) and sea ice concentration (SIC) as surface boundary conditions for the AGCMs, the authors explore the range of direct responses across three different dynamical models, and the response across two resolutions for each model.

Previous coupled simulations have shown an importance of model resolution in their representation of Southern Ocean polynyas (e.g. Lockwood et al. 2021), however there is limited research isolating the model response (as opposed to formation combined with response), and understanding how this varies with model resolution. Therefore understanding the effect of model resolution on the atmospheric response to polynyas is a valuable field of study. Comparing the direct atmospheric response to a reanalysis response can also provide some insight on the pathways for a climate response to the WSP. Finally, the study of the reanalysis response itself, with the state-of-the-art reanalysis model in ERA5 is also valuable in understanding the real-world response to WSPs.

Summarising the above paragraph, in my opinion the three main valuable contributions of this research paper are as follows:

• Exploring how model resolution impacts the atmospheric response to the WSP.

• Understanding the atmospheric-only response, and how this compares to the reanalysis response to provide hints on the relative importance of the direct atmospheric response with respect to responses which include coupled feedbacks.

• Analysing the ERA5 reanalysis response to the WSP.

I also would like to recognise that although the results of this study are primarily not unexpected or conceptually new, I still consider it a valuable contribution to repeat methodologies with state-of-the art models. Additionally, this research extends previous studies with a multi-model, multi-resolution framework to provide a comprehensive study of the direct atmospheric response to the WSP.

We thank the reviewer for their summary, which highlights our contributions, while acknowledging they are not unexpected. As we emphasize in our response to reviewer #2, our findings remain speculation until they are actually computed and disseminated to the community. Also, they are not expected to all researchers, especially those who have suggested large atmospheric responses and coupled feedbacks in the polynya dynamics.

Indeed, our results are somewhat unexpected "negative" results, but they set a robust basis and a novel perspective on how to interpret the dynamics of the Weddell Sea polynya in the coupled system.

Suggested revisions:

In my opinion, the authors need to take care that the wording of conclusory remarks remains within the scope of this direct atmosphere-only response study.

For example, the authors' statement "...our results suggest that a 1974-size polynya (the largest observed) cannot generate a response that would project on the scale of the Weddell gyre." is too much of an extrapolation, the study is isolated to atmosphere-only dynamics, and so can not alone rule out wider responses which could arise from coupled feedbacks.

We have edited this statement as follows: "our results suggest that a 1974-size polynya (the largest observed) cannot generate a *first order atmospheric* response that would project on the scale of the Weddell gyre."

Similarly, I think the following statement should be reworded, its wording implies that the results of this study prohibit a large-scale interaction of the WSP with the climate, whereas in reality, the results only suggest that because the reanalysis results and atmosphere-only results are similar, that the influence of the coupled feedbacks on the atmosphere are small. The WSP may influence wider ocean circulation, which could in turn have climate impacts elsewhere, and this possibility cannot be ruled out by this study alone: "...the WSP may not interact with the climate on a large scale and may be too far south and within sea ice edge for coupled feedbacks to have a substantial impact on results".

We have edited this statement as follows: "...the WSP may not interact, *to first order through the atmosphere* with the climate on a large scale and may be too far south and within sea ice edge for coupled feedbacks to have a substantial impact on results".

Again in the statement "In addition, our results show that the response has little memory (i.e., local in time too) and vanishes rapidly with the polynya", the authors need to clarify that this means that there is little memory in the atmospheric-only system. Memory in a climate system is often established via ocean-atmosphere-sea ice feedbacks, which by design are not present in this study.

We have edited this statement as follows: "In addition, our results show that the response has little memory *in the atmospheric-only system* (i.e., local in time too) and vanishes rapidly with the polynya".

The statement on line 94, "The interpretation of coupled models is made difficult by the potential impact of ocean-atmosphere-sea ice feedbacks", needs to be justified. A reader might argue that a coupled model is easier to interpret, because it includes feedbacks which an atmosphere-only model does not, and the processes which drive these feedbacks exist in the real-world. Comments such as this, and as those quoted above, present the idea that these direct atmosphere-only simulations are a better alternative to coupled simulations, when in fact these different methodologies answer different scientific questions.

We have edited this statement as follows: "The interpretation of coupled models *and of the direct response to the WSP* is made difficult by the potential impact of ocean-atmosphere-sea ice feedbacks ".

The authors have discussed the fact that they are using ERA5 before the assimilation of satellite data, and commented on potential data quality issues. They have also cautioned the

reader that the ERA5 data is the result of one realisation. I would therefore strongly recommend that the authors utilise all ten members of the ERA5 reanalysis to improve their analysis of the response to the polynya in ERA5, and to provide uncertainty estimates on this response. ERA5 is likely weakly observationally constrained in the Southern Ocean, particularly in the pre-satellite era, which means that the underlying model may play a significant role in determining the reanalysis conditions in the area of interest.

We have edited our analysis and the manuscript as suggested (figure 8), using all 10 members. We find that this does not change our results, but makes them more reliable.

With regards to the statistical methodology, I would suggest that the authors consider employing multiple-hypothesis testing when applying the students t-test to potentially spatially correlated data (e.g. surface temperature and precipitation response). As described by Wilks (2006), this is a mistake that is made by standard in the field. In my opinion, at least acknowledging this potential statistical caveat is an important step in improving the statistical robustness in our community.

We thank the reviewer for bringing this caveat to our attention. We have redone the statistical analysis to include False Detection Rate to our t-test p-values for figures 2,3,4,5 and 8. This new analysis shows that the dynamic response (and that of the response aloft) is less significant than suggested by our original statistical analysis.

However, in our previous draft, we had already downplayed these responses on the basis that they were inconsistent across models and had suggested they were due to internal variability. The new analysis reinforces and formalizes further our key message.

The paper is generally well written and presented, but could also benefit from a proofread. I have included below a list of some of the minor errors which I noticed, but I did not have the time to go through the entire article from a spelling and grammar point of view:

Spelling mistake in line 16 "isolate" should be "isolates".

Spelling mistake in line 68 "impact" should be "impacts".

Spelling mistake in line 112 "your" should be "our".

Line 121 "...thus, has limitations" the reason for limitations needs to be clarified a little.

Line 180 "limiting".

Authors should decide to use "sea ice" or "sea-ice" but not both.

We thank the reviewer for pointing these mistakes out, we have made corrections and further proofed the manuscript.

Finally, I would like to thank the authors for their contribution, which is a very interesting research topic.

Kind regards, Tarkan A Bilge

Lockwood, J.W., Dufour, C.O., Griffies, S.M., and Winton, M.: On the role of the Antarctic Slope Front on the occurrence of the Weddell Sea polynya under climate change, J. Climate, 34, 1–56, 2021.

Wilks, D. S., 2006: On "Field Significance" and the False Discovery Rate. _J. Appl. Meteor. Climatol._, 45, 1181–1189, [https://doi.org/10.1175/JAM2404.1] (https://doi.org/10.1175/JAM2404.1)

Review 2:

I appreciate the authors' effort to address some of the issues mentioned in the earlier manuscript review. Most of the authors' rewriting efforts greatly help better understand the results and discussion section of the paper. However, that alone does not recommend publication, considering that, as the authors themselves state, "the results are somewhat expected." The authors have presented no new analysis other than what was in the original submission.

In our earlier response to reviewers, we did point out that our "results are somewhat expected". However, the reviewer did not account for other points made in our response, where we justify the study's contribution to the field, both as part of the review and in our manuscript edits. Additionally, we hope and strongly believe that publication should not be based on results being expected or surprising. Many researchers start out a research project with expectations about their results, which is often a good practice. We expected that the direct atmospheric response to the polynya was small, and we did find this. It does not make our work irrelevant or uninteresting to the community and our work still very much delivers a novel result.

Prior to our study, the global three-dimensional atmospheric response to the polynya had not been assessed in detail. As we have highlighted in our introduction, previous studies used a two-dimensional domain, regional scale domain, or coupled models where interpretations are ambiguous due to the effect of secondary feedbacks with the ocean.

We have, for the first time, established the direct atmospheric response to the polynya. We have done it in a robust manner by considering 6 set-ups (3 models at 2 different resolution each) to avoid the usual uncertainties in our field of results relying on one single model. We have analysed the response and compared it to previous simulations and inferences from reanalysis to the extent it was possible, which we have extended upon in this latest version of the manuscript, after suggestions from reviewer #1.

The response is highly localized in space and time, where the small magnitudes of the responses do not lend themselves to generate a complex story. Additionally, we find that the limited dynamic response to the polynya that we downplayed because it was not robust across our set of models is not statistically significant according to the False Detection Rate analysis suggested by reviewer #1, reinforcing our conclusions. We nonetheless explored the link with the southern jet stream and SAM which have been suggested in past studies, and only found a tenuous link. Finally, our results suggest that inferences about the polynya dynamics from coupled models with overestimated polynya size should be taken with caution.

This also emphasizes that, while the results were expected to both us and reviewer #2, they would be unexpected to researchers who have argued that the polynya has a large hemispheric scale impact on the atmospheric circulation and interpreted the polynya as part

of a coupled mode (e.g., Diao et al., 2022; Weijer et al., 2017). We could have adopted their perspective and labelled our results as unexpected based upon these studies.

Our results should be published so future studies of the polynya dynamics can account for coupled interactions and feedbacks and the size of the polynya. Hence the importance of 'negative results' such as those presented in our study. Expected or not, the direct response to the polynya remains speculative until it is actually computed and disseminated to the community.

The manuscript does not have enough analysis to present a well-rounded story that would do justice to the goal of the manuscript. The authors do not save any of the outputs related to cloud characteristics, which is strange considering they wanted to study the atmospheric response. Previous studies that look at the atmospheric response to a Weddell Sea Polynya have found that most of the atmospheric response is limited to the vicinity of the open ocean polynya in terms of cloud and precipitation changes. Thus, the authors would have wanted to save the atmospheric variables that quantify the cloud characteristics and analyse the difference in cloud and precipitation characteristics with changing resolution.

We understand that the reviewer is particularly interested in clouds and radiations in the vicinity of the polynya, but when we started the project, our concerns were much wider. We had questions about the atmospheric circulation on a global scale, justly based upon the previous literature. Unlike the suggestion made by the reviewer, there wasn't a consensus about a localized response to the WSP. In fact, previous studies have suggested impact as far as the tropics (e.g., Diao et al., 2022, Chang et al., 2020, Kaufman et al., 2020). Again, this may reflect that reviewer #2 have their view on what the atmospheric response to the polynya is, but this view is not what appears in the literature (including recent papers).

Upon setting up our models, we had to make choice about which variables to save, at which frequency, under the constraints of limited storage (we nonetheless reached over 20 TB of data across all six models). A posteriori, with a different perspective one might want to rerun all the simulations; however, this is not possible within the resource and time limitations of our project. Our manuscript already presents changes in surface fluxes including precipitation from the submission and also provided further radiative fluxes changes in response to the earlier reviewer's comment that support said conclusions.

Furthermore, it is justifiable that at least one of the simulations should have a higher spatial resolution of ~0.1°.

We argue below that the request from the reviewer is unreasonable and unjustified.

It may be "justifiable that at least one of the simulations should have a higher spatial resolution of ~0.1°", but the reviewer does not provide such justification, here nor in the previous review.

We considered what could justify doing so under the constraints of limited resources. The most obvious is that the science question dictates simulations at 0.1° to resolve the process we investigate (for example, simulation of mesoscale front embedded in a synoptic eddy). We carried out simulations with a range of resolution from 0.22° (25 km) to 2.7° (300 km), which

show, for all purposes of the study, the same features, except for an increase in the magnitude of the response, which we explore. There are no indications that something drastically different would happen at 0.1°, until perhaps the convection scale is reached but this is much higher than 0.1°, and so the reviewer's focus on 0.1° remains unclear).

Our range of resolution covers the overwhelming majority of the resolution range currently used in global three-dimensional simulations in the atmospheric and coupled modelling (our range covers the entirety of the CMIP6 atmospheric models and a vast majority of high-res simulations such as those of PRIMAVERA project).

It is always possible to ask for more resolution (until the Kolmogorov scale is reached) and one could always speculate that something unexpected would happen at higher resolution. We do not think this is the right approach to planning research. The reviewer's request is a rather blunt statement that could apply to 100% of the published atmospheric modelling literature. Such reasoning could equally apply to observations, one could always ask for a denser array of instruments.

Simulations at 0.1° resolution on the global scale are extremely expensive, and we simply do not have the means to carry such an experiment. Such resolutions begin to bridge the gap between climate simulations and weather forecasts, such as the MPI ICON model, which is not the standard for the nature of our, or any related literatures study.

Working with limited resources we had to make choices to use optimally our resources. We choose to invest in a range of models and resolution that would establish the robustness of our results and cover the range of resolutions that the climate community uses (note that our 25 km runs correspond to the highest atmospheric resolution submitted to CMIP6).

Finally, to emphasize, computer resources roughly scale as the cube of the resolution (doubling the horizontal resolution quadruples the number of grid points, and halves the time steps, i.e. 8 times more resources). A 0.1° resolution runs would use $(2.5)^3=15.6$ time more resources than our 0.25° runs. In other words, a 0.1° run would have used about *5 times the entirety of the resources* invested in our 6 simulations.

---

## Author Response (AR3)

We thank the reviewer for their helpful feedback on our manuscript. We address the reviewer's technical suggestions below. We have included clarifications and changes to the manuscript.

I would like to thank the authors for their thorough response to my first review, and I am glad to see that they have incorporated all my previous suggestions. The authors have reworded and clarified statements regarding the impact of the study, applied a method to control the False Detection Rate, and utilised all ten members of ERA5 reanalysis. I would also like to thank the authors' for their further explanation of the significance of their study in the context of other recent research.

I now just have a couple of minor suggestions for the authors in light of their revisions:

- It would be good to include the reference for the Benjamini-Hochberg method in your bibliography.

  We have updated the bibliography for the above reference.

- The author has written "...by t-test and False Discovery Rate" in a number of places, it would be more accurate to replace "False Discovery Rate" in this context by "The Benjamini-Hochberg procedure". Technically, the False Discovery Rate is the rate at which statistical tests falsely appear significant, and the Benjamini-Hochberg procedure is a method to control the FDR (i.e. it is a "False Discovery Rate controlling procedure").

  We have updated the manuscript with the suggested changes.

- Is there a version number that accompanies HadGEM3? I thought that HadGEM3 was the name of the whole model family - perhaps you could check this, I could be wrong here. For example, perhaps it is called HadGEM3-AO. The reference given (Williams et al. 2017) doesn't seem to be in your bibliography.

  As mentioned on line 151, we use the GA7.1 model version (Walters et al. 2017). Thank you for pointing out the Williams reference, this is no longer needed in this version of the manuscript and has been removed. The Walters reference has been added to the bibliography.

- In my opinion, the point in line 94 "The interpretation of coupled models and of the direct response to the WSP is made difficult..." should be reworded to "Interpreting the direct response of the WSP in coupled models is made difficult..." or similar - at the moment it does not quite work grammatically.

  We have updated the manuscript with the suggested changes.

- Could you double-check the bibliography, I have noticed a couple of missing entries (above) but also some of the entries are incomplete, e.g. "Timmermann, R., P. Lemke, and C. Kottmeier, 1999: Formation and Maintenance of a Polynya in the Weddell Sea." has no journal name.

  We have updated the bibliography for the above references and proofed the references.

- Typo in Figure 6 caption; "solid".

  This has now been corrected in the manuscript.